# CDK1 and CDK2 regulate NICD1 turnover and the periodicity of the segmentation clock

Francesca Anna Carrieri[1], Philip J Murray[2], Dimitrinka Ditsova[1], Margaret Ashley Ferris[3], Paul Davies[4] & Jacqueline Kim Dale[1,*] (ID)

## Abstract

**All vertebrates share a segmented body axis. Segments form from the rostral end of the presomitic mesoderm (PSM) with a periodicity that is regulated by the segmentation clock. The segmentation clock is a molecular oscillator that exhibits dynamic clock gene expression across the PSM with a periodicity that matches somite formation. Notch signalling is crucial to this process. Altering Notch intracellular domain (NICD) stability affects both the clock period and somite size. However, the mechanism by which NICD stability is regulated in this context is unclear. We identified a highly conserved site crucial for NICD recognition by the SCF E3 ligase, which targets NICD for degradation. We demonstrate both CDK1 and CDK2 can phosphorylate NICD in the domain where this crucial residue lies and that NICD levels vary in a cell cycle-dependent manner. Inhibiting CDK1 or CDK2 activity increases NICD levels both *in vitro* and *in vivo*, leading to a delay of clock gene oscillations and an increase in somite size.**

**Keywords** cell cycle; FBXW7; Notch; phosphorylation; somitogenesis
**Subject Categories** Development & Differentiation; Post-translational Modifications, Proteolysis & Proteomics; Signal Transduction

See also: **KM Braunreiter & SE Cole** (July 2019)

## Introduction

Segmentation, a process which occurs early during vertebrate body plan formation, generates repeated segments (or somites) that later give rise to the vertebral column, most skeletal musculature and dermis [1,2].

During somitogenesis, pairs of somites bud off the rostral end of the unsegmented presomitic mesoderm (PSM) with a periodicity that is species specific. The periodicity of segment formation is regulated by a molecular oscillator, known as the somitogenesis clock, which drives oscillatory gene expression within the PSM tissue from which somites are derived [2–4].

The genes that exhibit oscillatory PSM expression are termed clock genes and are targets of the Notch, Wnt and FGF pathways [5,6]. Aberrant somitogenesis leads to severe segmentation and skeletal defects [7]. In humans, defects in segmentation lead to congenital scoliosis (CS), with an infant mortality rate of 50% that comprises many vertebral skeletal and muscular pathologies, including the family of spondylocostal dysostoses (SCD). For CS, while the aetiology is unclear, linkage analyses have shown mutations in six genes (DLL3, MESP2, LFNG, HES7, TBX6 and RIPPLY2) lead to familial forms of SCD [8]. Significantly, four of these are components of the Notch pathway (DLL3, MESP2, LFNG, HES7), which plays multiple roles during segmentation. Notch is crucial to the segmentation process in mice, since in the absence of Notch signalling, the segmentation clock stops and no somites form [9].

On a single cell level in the PSM, oscillatory clock gene expression is established through positive and negative feedback loops of unstable clock gene products, which potentiate or inhibit the pathway that activates them. Synchronization of clock gene oscillations between neighbouring cells is reliant on Notch signalling [10–13]. Mathematical models predict the period of clock gene oscillations can be approximated as a sum of the delays involved in transcription, splicing, translation and transport of clock gene products, and in particular through the regulation of the half-lives of both mRNA and protein of unstable regulators [14–17]. Progress has been made in demonstrating the role of transcription and splicing delays in setting the clock period. However, with the respect to the half-life of clock components this has been shown to play an important role in clock function [18–20], but little experimental work investigating whether stability of clock components affects clock period has been performed.

Most studies addressing the molecular mechanisms regulating the periodicity of clock gene oscillations have focused on the role of Notch signalling components [10,19,21–25].

Notch is one of the major highly conserved signalling pathways that regulate cell–cell communication, which involves gene regulation mechanisms that control multiple processes during development and adult life [26–33].

1 Division of Cell and Developmental Biology, School of Life Sciences, University of Dundee, Dundee, UK
2 Department of Mathematics, University of Dundee, Dundee, UK
3 Department of Pediatrics, Washington University School of Medicine, Saint Louis, MO, USA
4 Medical Research Council Protein Phosphorylation and Ubiquitylation Unit, School of Life Sciences, University of Dundee, Dundee, UK
 *Corresponding author. Tel: +44 01382 386290; E-mail: j.k.dale@dundee.ac.uk

Upon extracellular ligand binding, Notch transmembrane receptors are cleaved, releasing the intracellular domain (NICD) that translocates to the nucleus to regulate expression of specific developmental gene cohorts [32–34]. NICD is highly labile, and phosphorylation-dependent turnover acts to restrict Notch signalling [35–37].

Most canonical Notch activity relies on this regulation of NICD turnover. Moreover, aberrant NICD turnover contributes to numerous cancers and diseases [26,30,38–43]. Despite the multiple impacts of NICD turnover in both development and disease, the molecular mechanism regulating this turnover remains largely uncharacterized. The stability of NICD and therefore duration of the Notch signal is regulated by phosphorylation of the C-Terminal PEST domain which leads to subsequent recruitment of FBXW7, F-Box and WD Repeat Domain Containing 7 (a key component of the SCF$^{Sel10/FBXW7}$ E3 ubiquitin ligase complex) [35–38,44–49]. Ultimately, this leads NICD to ubiquitylation and proteasomal degradation [44,50–53].

However, the molecular details of NICD degradation mediated by FBXW7 are not well understood.

A recent study combining experimental and computational biology demonstrated changes in NICD stability dramatically affects the chick and mouse somitogenesis clock period, which in turn affects somite size. In this study, a pharmacological approach was used to demonstrate that culturing chick/mouse PSM explants with broad specificity inhibitors leads to elevated levels and a prolonged NICD half-life and phase shifted clock oscillation patterns at a tissue level, leading to larger segments. Furthermore, reducing NICD production in this assay rescues these effects [21]. These results imply potential coupling between NICD degradation and the segmentation clock. However, the molecular mechanism of action of the inhibitors was not explored and leaves open the question of whether this coupling is a general or conserved mechanism.

In this manuscript, we identify the phosphorylated residues within human NICD. We demonstrate that purified recombinant Cyclin-dependent kinase 1 (CDK1) and Cyclin-dependent kinase 2 (CDK2) phosphorylate NICD within the PEST domain. A point mutation affecting a conserved serine residue within this CDK substrate domain of the NICD PEST motif prevents NICD interaction with endogenous FBXW7. Strikingly, we show that NICD levels fluctuate in a cell cycle-dependent manner correlating inversely with high levels of CDK1/2 activity. Lastly, we demonstrate, using a variety of assays, that inhibition of CDK1 or CDK2 activity leads to increased levels of NICD *in vitro* and *in vivo* and delays the mouse somitogenesis clock and somite formation, leading to larger somites.

Using a mathematical model, we show that the experimental observations made in cell lines and PSM tissue can be explained in a single theoretical framework that couples the cell cycle to NICD degradation.

# Results

## Roscovitine, DRB and XAV939 increase NICD levels in HEK293, iPS, mES and IMR90 cells

A study using broad-range inhibitors demonstrated that the stability and turnover of NICD is linked to the regulation of the pace of the segmentation clock across the PSM in chick and mice embryos [21]. However, this study did not define the molecular mechanism of action of the inhibitors.

It has been reported that direct phosphorylation of NICD in its PEST domain enhances its turnover and thus degradation [35–37]. In order to identify which kinases are involved in NICD phosphorylation and which residues in the NICD PEST domain are phosphorylated rendering NICD susceptible to degradation, we employed a cellular model due to the limiting quantity of material available using embryonic cell lysates.

First, we used the same inhibitors as Wiedermann *et al* [21] and thus investigated if Roscovitine, DRB and XAV939 elicit the same effect upon NICD levels in a variety of cell culture models, namely HEK293 (human embryonic kidney), iPS (induced pluripotent stem cells), mES (mouse embryonic stem cells) and IMR90 (human Caucasian fetal lung fibroblast) cells.

Roscovitine is a small molecule belonging to the family of purines. It inhibits Cyclin-dependent kinases (CDKs) through direct competition with ATP for binding at the ATP-binding site of CDKs [54,55]. DRB (5,6-dichloro-1-β-D-ribofuranosylbenzimidazole) also inhibits CDKs, particularly CDK7 and 9 [56,57]. XAV939 is a Wnt inhibitor that stimulates β-catenin degradation by stabilizing axin through inhibition of the enzymes tankyrase 1 and 2 [58].

HEK293, iPS, mES or IMR90 cells treated for 3 h with each of the three inhibitors at the same concentrations used in embryonic lysate studies [21] led to an increase in NICD levels compared to control cells cultured in the presence of DMSO (Figs 1A–D and EV1A–C). In control conditions, NICD was not easily detectable due to its very short half-life. Quantification of the density of Western blot bands in at least three independent experiments confirmed that the increase in NICD levels was statistically significant after treatment with Roscovitine, DRB and XAV939, as shown in Figs 1B and D, and EV1B. Two other inhibitors were used as positive and negative controls for the assay. LY411575 is a γ-secretase inhibitor that prevents Notch1 cleavage and thus inhibits activation of target gene expression [59,60]. As expected, LY411575 treatment significantly reduced NICD levels (Figs 1A–D and EV1A–C). Phosphorylation of the C-Terminal PEST domain of NICD leads to recruitment of FBXW7 and thus to NICD ubiquitylation and proteasomal degradation [35–38,44,46–48]. When E3 ligase activity is reduced with the NEDDylation inhibitor MLN4924 [61], NICD levels increase, since NICD degradation is stopped in the presence of this compound (Figs 1A–D and EV1A–C).

Interestingly, we were able to detect two distinct bands by Western blot with the NICD antibody, particularly when cells were treated with MLN4924. We hypothesized this reflected the presence of non-phospho and phospho-NICD species. To test this hypothesis, we treated MLN4924-treated lysates with λ phosphatase, which abrogated the appearance of the higher band by Western blot with the NICD antibody (Fig 1A and C). These data demonstrate that the higher band detected corresponds to a phosphorylated form of NICD. It is noteworthy that quantification of NICD levels following MLN4924 and λ phosphatase treatment reveals that they are not significantly different to NICD levels in DMSO-treated cells (Fig 1B and D), albeit that NICD levels can vary considerably in DMSO-treated lysates. This suggests an additional scientific possibility, which is that when all phosphorylation modifications are removed upon λ phosphatase treatment, NICD is also very unstable and degraded by a mechanism other than SCF E3 ligase, indicating that some phosphorylation events may actually stabilize NICD.

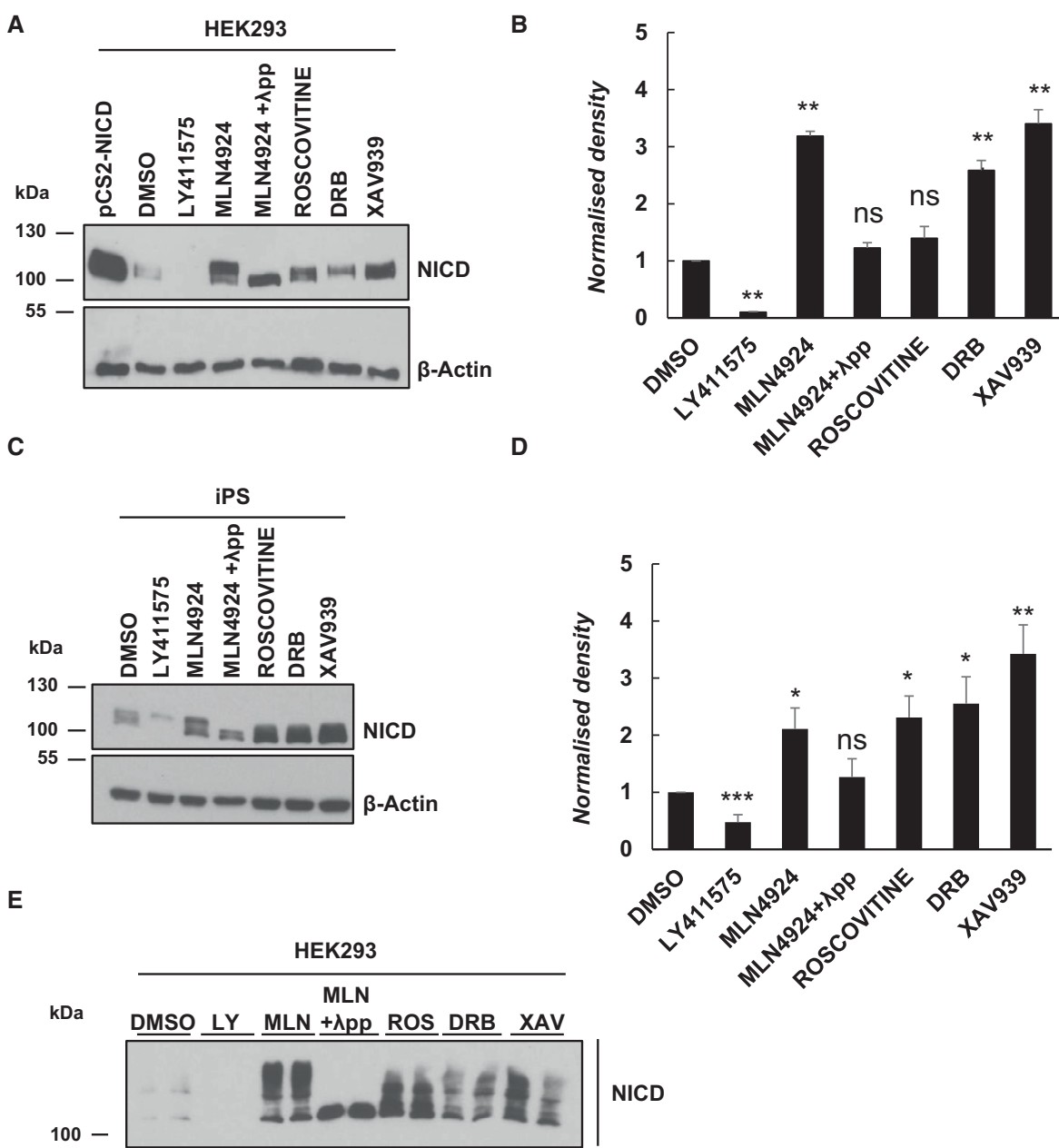

**Figure 1. Endogenous NICD levels increase in HEK293 cells following treatment with Roscovitine, DRB and XAV939.**

A  HEK293 cells were treated for 3 h with 150 nM of LY411575, 1 μM of MLN4924, 10 μM of Roscovitine, 10 μM DRB or 10 μM XAV939. DMSO served as vehicle control. Transfection with pCS2-NICD vector served as positive control. Western blot analysis reveals that NICD levels were increased upon treatment with Roscovitine, DRB, XAV939 or MLN4924. NICD is undetectable following LY411575 treatment. NICD antibody detects a doublet following inhibitor treatment, and the top band disappears following λ phosphatase treatment indicating that this top band is a phosphorylated isoform. β-Actin served as loading control.

B  Quantification of the density of Western blot bands in (A) using ImageJ software. Data are expressed as fold changes compared to DMSO treatment. All data represent the mean ± SEM from three independent experiments. One-way ANOVA analysis, followed by Dunnett's test, was performed with **$P \leq 0.01$ and ns = not significant.

C  iPS cells were treated for 3 h with 150 nM of LY411575, 1 μM of MLN4924, 10 μM of Roscovitine, 10 μM DRB or 10 μM XAV939. DMSO has been used as vehicle control. Western blot analysis reveals that NICD levels were increased upon treatment with Roscovitine, DRB, XAV939 or MLN4924. NICD is undetectable following LY411575 treatment. NICD antibody detects a doublet following Inhibitor treatment and the top band disappears following λ phosphatase treatment indicative that this top band is a phosphorylated isoform. β-Actin served as loading control.

D  Quantification of the density of Western blot bands in (C) using ImageJ software. Data are expressed as fold changes compared to DMSO treatment. All data represent the mean ± SEM from three independent experiments. One-way ANOVA analysis, followed by Dunnett's test, was performed with *$P \leq 0.05$, **$P \leq 0.01$, ***$P \leq 0.001$ and ns = not significant.

E  HEK293 cells were treated with the same inhibitors as described in (A). NICD phosphorylation status was analysed by a Phos-tag assay. DMSO served as vehicle control. NICD phosphorylation profile varies following Roscovitine, DRB, XAV939 or MLN4924 treatment. Following λ phosphatase treatment, none of the high molecular weight bands are visible indicating they are all phosphorylated isoforms.

In order to determine whether the increased levels of NICD were due to increased NICD production and/or increased NICD stability, we exposed HEK293 cells to LY411575 treatment for the last hour of culture, thereby inhibiting the production of new NICD. Under control conditions with LY411575 treatment in the last hour, NICD levels are very low (Appendix Fig S1A, lane 1). However, despite LY411575 treatment in the last hour, cells cultured in the presence of small molecule inhibitors showed increased levels of NICD compared to the control, indicating that the increase in NICD levels is not due to increased NICD production, but to an increased stability (Appendix Fig S1A).

Taken together, these results show that exposure to this group of inhibitors leads to increased levels of NICD in a variety of cell lines, in the same way that they do in the mouse and chicken PSM tissue, suggesting they regulate a conserved mechanism leading to increased NICD levels and reduced NICD turnover.

### Phos-tag analysis following Roscovitine, DRB and XAV939 treatment reveals a variety of NICD phospho-species

In order to investigate whether this selection of small molecule inhibitors has different effects on NICD phosphorylation, we treated HEK293 cells with the inhibitors and performed a Phos-tag assay [62]. This modified Western blot assay allows simultaneous visualization of different phosphorylated isoforms of a given protein of interest, as a result of their different migration speeds. Following MLN4924 treatment, a variety of bands indicative of different phospho-species of NICD was observed (Fig 1E and Appendix Fig S1B). Given that very few bands are present in the control sample (DMSO, Fig 1E), the bands detected after MLN4924 treatment are likely to be very labile isoforms of the NICD peptide, which are rapidly degraded in the DMSO sample. In contrast, when whole-cell lysate was treated with both MLN4924 and λ phosphatase, only one band, of the lowest molecular weight, was detectable, further supporting the notion that the ladder of bands obtained upon MLN4924 treatment reflects a variety of unstable phosphorylated NICD isoforms, which is completely depleted in the presence of λ phosphatase (Fig 1E). As expected from data showing NICD levels are increased after inhibitor treatment (Fig 1A–D), all three inhibitors Roscovitine, DRB and XAV939 cause a noticeable increase in the number and intensity of bands compared to control cells. However, compared to the effect seen with MLN4924, Phos-tag technology reveals Roscovitine, DRB and XAV939 have a reduced number of phospho-bands as compared to MLN4924, indicative of the fact these inhibitors act to reduce NICD phosphorylation. Moreover, each of the inhibitors presents a distinct profile of NICD phospho-species. These data suggest that NICD is targeted by several kinases and/or phosphorylation events, which are differentially sensitive to these inhibitors (Fig 1E and Appendix Fig S1B). In each case, however, the highest molecular bands were no longer visible. Concomitant treatment with various inhibitors together with MLN4924 followed by Phos-tag analysis demonstrates the inhibitors are actually preventing the production of the uppermost phosphorylated band of NICD that accumulates after MLN4924 inhibition alone (Appendix Fig S1B). It is possible that the multiple phosphorylation bands are indicative of unique events, some of which may reflect priming phosphorylation events that facilitate or increase the efficiency of secondary phosphorylation events, which then act as

phospho-degron signals to recruit E3 ligases that target NICD for degradation.

### NICD-FBXW7 interact at the endogenous levels in HEK293 cells, in a phosphorylation-dependent manner

The involvement of the F-box protein component of the SCF E3 ligase complex, FBXW7, in NICD degradation has been previously reported [46–48]. However, to date, the NICD-FBXW7 interaction has only been shown in overexpressed systems [36,38,46–48,63,64]. Thus, we examined the binding of NICD to FBXW7 using co-immunoprecipitation analysis at the endogenous level.

FBXW7 was immunoprecipitated from HEK293 cells treated with DMSO or MLN4924 for 3 h, and extracts were probed with NICD antibody. NICD directly binds to FBXW7 (Fig 2A). After MLN4924 treatment, an inhibitor of Cullin1 neddylation, the amount of NICD bound to FBXW7 was significantly higher compared to control cells and this was particularly evident with the higher molecular weight isoform of NICD (Fig 2A and B), confirming again NICD-FBXW7 interaction is phosphorylation-dependent.

In order to determine whether the change in the NICD phosphorylation profile observed after treatment with the CDK inhibitors reduced the NICD-FBXW7 interaction, we performed the same co-immunoprecipitation assay after CDK inhibitor treatment. In order to maximize the amount of NICD immunoprecipitated with FBXW7, cells were treated with MLN4924 (to prevent NICD degradation) in the presence or the absence of the CDK inhibitors. A significantly reduced interaction between NICD and FBXW7 was observed after treating HEK293 cells with Roscovitine or DRB for 3 h (Fig 2C and Appendix Fig S2A). Statistical analyses, carried out on the density of Western blot bands after immunoprecipitation, confirmed a significant reduction in the interaction between NICD and FBXW7 following either Roscovitine or DRB treatment (Fig 2D and Appendix Fig S2B).

Taken together, these data demonstrate, for the first time, that NICD interacts with FBXW7 at endogenous levels in HEK293 cells, and this interaction is dependent on phosphorylation.

To further validate the involvement of FBXW7 in endogenous NICD turnover, we conducted a siRNA-mediated depletion of FBXW7 in HEK293 cells (Appendix Fig S2E). siRNA treatment efficiently depleted FBXW7 protein levels and led to an increase in levels of the FBXW7 target protein Cyclin E. FBXW7 depletion also resulted in increased levels of NICD and, in particular, an accumulation of the phosphorylated form of NICD (Appendix Fig S2E).

### Serine 2513 is essential for the NICD-FBXW7 interaction

We utilized Mass Spectrometry as an unbiased approach to identify which NICD residues are phosphorylated in HEK293 cells transiently transfected with human NICD-GFP, followed by immunoprecipitation of NICD-GFP. Gel slices were processed and submitted to MS analysis. We identified 15 phospho-sites on exogenous hNICD, highlighted in green in Fig 3A. To investigate the relevance of those phosphorylation sites in NICD turnover, we screened those located within the PEST domain (such as S2527), and others based on the FBXW7 phospho-degron motif (such as S2205, S2513, S2516, S2538) which is known to be [RK] S/T P [RK] X S/T/E/D, where X is any amino acid and RK is any amino acid except arginine (R) or

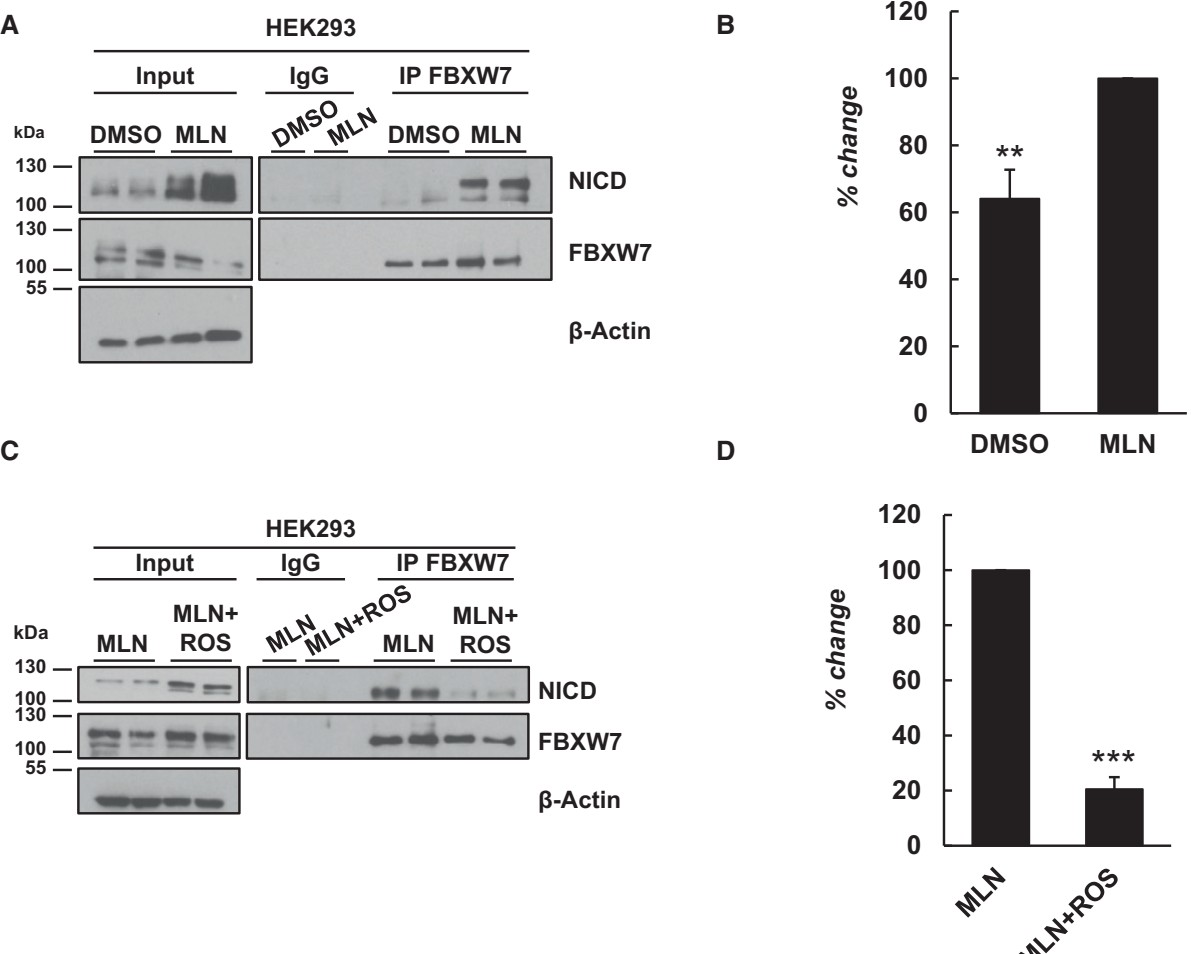

**Figure 2. NICD and FBXW7 interact directly at endogenous levels.**

A NICD interaction with FBXW7 at endogenous levels in HEK293 cells. 500 μg of HEK293 cell lysates treated with DMSO or MLN4924 was subjected to immunoprecipitation using FBXW7 antibody, or IgG antibody as negative control, and precipitated material was analysed by Western blot using NICD antibody. Western blot with FBXW7 antibody served as loading control for immunoprecipitation efficiency. 10% of cell lysate before immunoprecipitation was used as input control and β-Actin served as loading control.

B Quantification of the density of Western blot bands in (A) performed by ImageJ software. Data are expressed as percentage changes compared to MLN4924. All data represent the mean ± SEM from three independent experiments. Student's *t*-test was used to determine *P* values, with **$P \leq 0.01$.

C Roscovitine treatment reduced the NICD-FBXW7 interaction. 500 μg of HEK293 cell lysates treated with MLN4924 or MLN4924 together with Roscovitine was subjected to immunoprecipitation using FBXW7 antibody, or IgG antibody as negative control, and precipitated material was analysed by Western blot using NICD antibody. Western blot with FBXW7 antibody served as loading control for immunoprecipitation efficiency. 10% of cell lysate before immunoprecipitation was used as input control and β-actin served as loading control.

D Quantification of the density of Western blot bands in (C) performed by ImageJ software. Data are expressed as percentage changes compared to MLN4924-treated samples. All data represent the mean ± SEM from three independent experiments. Student's *t*-test analysis was performed, with ***$P \leq 0.001$.

lysine (K) [65]. CDKs are proline-directed kinases, so we also investigated the relevance of proline residues adjacent to the consensus motif shared by most FBXW7 in NICD PEST domain.

Previous deletion studies have suggested some of these sites may play a role in regulation of NICD turnover [36,63]. Thus, we generated eight NICD constructs each carrying either a serine to alanine, threonine to valine or proline to leucine/arginine point mutations in an identified site. Following transient transfection of HEK293 cells with wild-type or mutated peptides, we performed immunoprecipitation using GFP-conjugated beads, to evaluate peptide binding efficiency with endogenous FBXW7. We found 7 of 8 of these

individual mutations (serine 2205, 2516, 2527 and 2538, threonine 2511 and proline 2512 and 2514) did not affect the NICD-FBXW7 interaction (Fig 3B, Appendix Fig S2F and G), at least in HEK293 cells. However, mutating serine 2513 to alanine, to render this residue non-phosphorylatable, completely abolished the NICD-FBXW7 interaction in this assay (Fig 3B). Cells transfected with the double-mutant S2513A/S2516A also showed a dramatic loss of the NICD-FBXW7 interaction. This did not reflect a reduction in the level of immunoprecipitated GFP (Fig 3B).

Thus, our data suggest that only serine 2513, of those we have tested, is the key NICD phosphorylation site required for interaction

## A

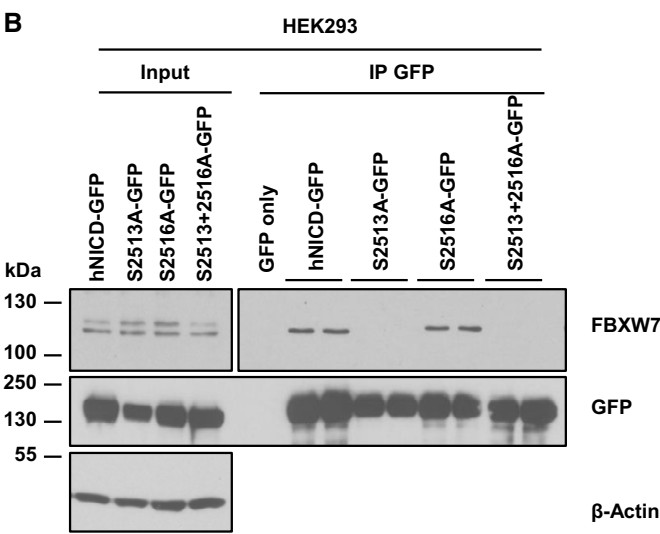

```
        1760        1770        1780        1790        1800
GCG VLLSRKR RRQHGQLWFP EGFKVSEASK KKRREPLGED SVGLKPLKNA
        1810        1820        1830        1840        1850
SDGALMDDNQ NEWGDEDLET KKFRFEEPVV LPDLDDQTDH RQWTQQHLDA
        1860        1870        1880        1890        1900
ADLRMSAMAP TPPQGEVDAD CMDVNVRGPD GFTPLMIASC SGGGLETGNS
        1910        1920        1930        1940        1950
EEEEDAPAVI SDFIYQGASL HNQTDRTGET ALHLAARYSR SDAAKRLLEA
        1960        1970        1980        1990        2000
SADANIQDNM GRTPLHAAVS ADAQGVFQIL IRRNRATDLDA RMHDGTTPLI
        2010        2020        2030        2040        2050
LAARLAVEGM LEDLINSHAD VNAVDDLGKS ALHWAAAVNN VDAAVVLLKN
        2060        2070        2080        2090        2100
GANKDMQNNR EETPLFLAAR EGSYETAKVL LDHFANRDIT DHMDRLPRDI
        2110        2120        2130        2140        2150
AQERMHHDIV RLLDEYNLVR SPQLHGAPLG GTPTLSPPLC SPNGYLGSLK
        2160        2170        2180        2190        2200
PGVQGKKVRK PSSKGLACGS KEAKDLKARR KKSQDGKGCL LDSSGMLSPV
        2210        2220        2230        2240        2250
DSLESPHGYL SDVASPPLLP SPFQQSPSVP LNHLPGMPDT HLGIGHLNVA
        2260        2270        2280        2290        2300
AKPEMAALGG GGRLAFETGP PRLSHLPVAS GTSTVLGSSS GGALNFTVGG
        2310        2320        2330        2340        2350
STSLNGQCEW LSRLQSGMVP NQYNPLRGSV APGPLSTQAP SLQHGMVGPL
        2360        2370        2380        2390        2400
HSSLAASALS QMMSYQGLPS TRLATQPHLV QTQQVQPQNL QMQQQNLQPA
        2410        2420        2430        2440        2450
NIQQQQSLQP PPPPPQPHLG VSSAASGHLG RSFLSGEPSQ ADVQPLGPSS
        2460        2470        2480        2490        2500
LAVHTILPQE SPALPTSLPS SLVPPVTAAQ FLTPPSQHSY SSPVDNTPSH
        2510        2520        2530        2540        2550
QLQVPEHPFL TPSPESPDQW SSSSPHSNVS DWSEGVSSPP TSMQSQIARI

PEAFK
```

## B

**HEK293**

Input | IP GFP

hNICD-GFP · S2513A-GFP · S2516A-GFP · S2513+2516A-GFP · GFP only · hNICD-GFP · S2513A-GFP · S2516A-GFP · S2513+2516A-GFP

kDa

130 —  FBXW7
100 —

250 —  GFP
130 —

55 —

β-Actin

#### Figure 3. Mass spectrometry analysis of phosphorylated residues in hNICD.

A LC-MS-MS analysis of in-gel-digested HEK293 cells transfected with hNICD-GFP and subjected to immunoprecipitation using GFP antibody, identified multiple phosphorylation sites in NICD, highlighted in green.

B Phosphorylation of serine 2513, but not serine 2516, appears to be essential for the NICD-FBXW7 interaction. hNICD-GFP phospho-mutant peptides encoding non-phosphorylatable residues at S2513 and/or 2516 (serine to alanine) were expressed in HEK293 cells. The exogenously expressed protein was subsequently immunoprecipitated with anti-GFP antibody, and precipitated material was analysed by Western blot using FBXW7 antibody. Wild-type hNICD-GFP and GFP only vectors were included as positive and negative controls, respectively. Western blot using GFP antibody served as immunoprecipitation efficiency control. β-Actin has been used as loading control for the input lanes.

with FBXW7, and thus potentially crucial for NICD stability and turnover.

### Cyclin-dependent Kinase (CDK) 1 and 2 phosphorylate NICD *in vitro*

Previous reports have proposed a number of potential kinases which may be involved in NICD phosphorylation and turnover, including CyclinC:CDK8 [36], CyclinC:CDKs [63], PKC [66], PIM [67] and GSK3β [35,37] kinases. The MRC-PPU Kinase Profiling Inhibitor Database (http://www.kinase-screen.mrc.ac.uk/kinase-inhibitors) (University of Dundee) indicates that, at the concentrations used in our assays, Roscovitine, in particular, is a potent inhibitor of CDK2, but a very weak inhibitor of CK1, GSK3β and more than 50 other kinases tested. In order to confirm results derived from the Kinase Profiling Inhibitor Database, we performed a kinase assay in collaboration with the MRC-PPU International Centre for Kinase Profiling. We tested three NICD phospho-peptides against the activity of a panel of different kinases including CDK1 and CDK2 (Table 1). Of seven kinases tested, 5 had no specific activity against any of the peptides. In contrast, CDK1 and CDK2 elicited a very high activity against Peptide 1, which contained serine residues 2513 and 2516, previously identified by mass spectrometry analysis to be phosphorylated in NICD, and in particular 2513, we have shown to be crucial for the NICD-FBXW7 interaction.

These results demonstrate that CDK1 and CDK2 phosphorylate the C-terminal region of NICD *in vitro*. Therefore, we decided to evaluate the contribution of these kinases to endogenous NICD turnover and FBXW7 interaction in the cell lines. CDK2 siRNA treatment in HEK293 cells efficiently depleted CDK2 protein levels, with no

#### Table 1. CDK1 and CDK2 exhibit specific activity against a NICD phospho-peptide by *in vitro* kinase assay.

| Specific activity (U/mg) | Peptide 1 | Peptide 2 | Peptide 3 | Control |
|---|---|---|---|---|
| Cdk1/CyclinA2 | **1035.43** | 35.21 | −7.86 | 1322 |
| Cdk1/CyclinB | **887.23** | 37.46 | 0.04 | 1566 |
| Cdk2/CyclinA | **79.7** | 0.5 | 0 | 117.7 |
| Cdk5/p35 | 47 | 0.8 | 0.5 | 1729.5 |
| Cdk7/MAT1/CyclinH | 0.1 | 0.2 | 0.1 | 21.6 |
| Cdk9/CyclinT1 | 2.7 | 0.6 | 0.3 | 66.4 |
| CK1α | 0 | 0 | 0.1 | 19.1 |
| GSKβ | 1 | 1 | 1.1 | 552.5 |

The figures in bold represent significant enzyme activity against the peptide as compared to activity of the respective enzyme on their control substrate in the last column.

Three NICD phospho-peptides were tested for the activity of seven different kinases (CDK1, CDK2, CDK5, CDK7, CDK9, CK1α and GSKβ). A known substrate was used as control for each kinase. The specific activity of each kinase is expressed in U/mg.

Peptide 1 contains serine residues 2513 and 2516 (HPFLTP**S**PE**S**PDQWSSSSPH).

Peptide 2 includes serine 2538 (NVSDWSEGVS**S**PPTSMQSQIA).

Peptide 3 encompasses serine residue 2141 (**G**TPTLSPPLC**S**PNGYLGSLKP).

Peptide 1: **S2513** and **S2516** HPFLTPSPESPDQWSSSSPH.

Peptide 2: **S2538** NVSDWSEGVSSPPTSMQSQIA.

Peptide 3: **S2141** GTPTLSPPLCSPNGYLGSLKP.

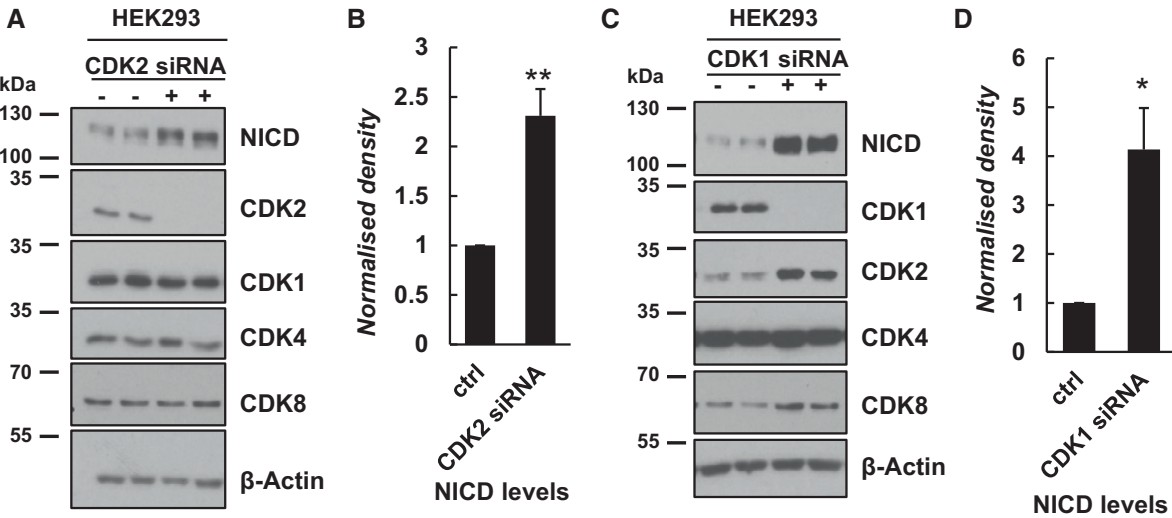

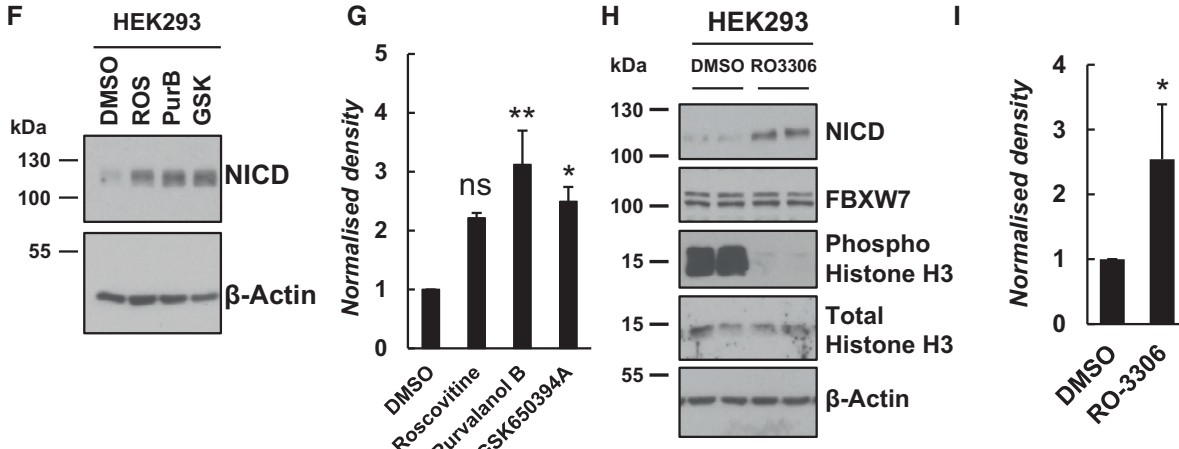

**Figure 4.**

**Figure 4.  Depletion or pharmacological inhibition of CDK1 or CDK2 increased endogenous levels of NICD in HEK293 cells.**

A   HEK293 cells were cultured for 48 h after transfection with plasmids encoding scrambled siRNA (−) or siRNA specific for CDK2 (+) followed by Western blot for NICD, CDK2, CDK1, CDK4 and CDK8. β-Actin served as loading control.

B   Quantification of the density of Western blot bands in (A) performed by ImageJ software. Data are expressed as fold changes compared to control (scrambled siRNA transfected) cell lysate. All data represent the mean ± SEM from three independent experiments. Student's *t*-test analysis was performed, with **$P ≤ 0.01$.

C   HEK293 cells were cultured for 48 h after transfection with plasmids encoding scrambled siRNA (−) or siRNA specific for CDK1 (+) followed by Western blot for NICD, CDK1, CDK2, CDK4 and CDK8. β-Actin has been used as loading control.

D   Quantification of the density of Western blot bands in (C) performed by ImageJ software. Data are expressed as fold changes compared to control (scrambled siRNA transfected) cell lysate. All data represent the mean ± SEM from three independent experiments. Student's *t*-test analysis was performed, with *$P ≤ 0.05$.

E   Analysis of the inhibitory activity of three highly selective CDK2 inhibitors against a panel of kinases—a selection of those tested is shown here. At both 1 and 10 μM, Roscovitine is able to inhibit more than 95% of CDK2 activity, but is far less effective against other kinases. Purvalanol B (0.1 and 1 μM) inhibits more than 94% of CDK2 activity. At both 1 and 10 μM, GSK650394A is able to inhibit more than 98% of CDK2 activity. Source: The Kinase Profiling Inhibitor Database (http://www.kinase-screen.mrc.ac.uk/kinase-inhibitors).

F   HEK293 cells were treated with three highly selective CDK2 inhibitors (10 μM of Roscovitine, 0.1 μM of Purvalanol B and 10 μM of GSK650394A) for 3 h. Endogenous levels of NICD were detected by Western blot. β-Actin served as loading control.

G   Quantification of the density of Western blot bands in (F) using ImageJ software. Data are expressed as fold changes compared to DMSO. All data represent the mean ± SEM from three independent experiments. One-way ANOVA analysis, followed by Dunnett's test, was performed, with *$P ≤ 0.05$, **$P ≤ 0.01$, and ns = not significant.

H   HEK293 cells were treated with RO-3306, a specific CDK1 inhibitor, for 3 h at 10 μM. Levels of NICD, FBXW7, phospho-histone H3 and histone H3 were detected by Western blot. β-Actin was used as loading control.

I   Quantification of the density of Western blot bands in (H) using ImageJ software. Data are expressed as fold changes compared to DMSO. All data represent the mean ± SEM from three independent experiments. Student's *t*-test analysis was performed, with *$P ≤ 0.05$.

effect on levels of CDK1, CDK4 or CDK8. Under these conditions, NICD levels were significantly increased compared to control scrambled siRNA-treated cells, indicative of reduced NICD turnover (Fig 4A and B).

We repeated this assay and depleted CDK1 by a siRNA-mediated approach in HEK293 cells. Under these conditions, we also observed an increase in NICD protein levels compared to the control (Fig 4C), and this increase was statistically significant (Fig 4D). Interestingly, we also detected elevated levels of CDK2 and CDK8 following CDK1 depletion, suggesting a possible compensation effect was occurring upon CDK1 knockdown. Nevertheless, this did not prevent the effect of loss of CDK1 upon NICD turnover.

CDK8 has previously been proposed as a potential kinase involved in NICD phosphorylation and turnover [36,63]. We monitored NICD levels following efficient siRNA-mediated depletion of CDK8 in HEK293 cells. This resulted in slightly elevated NICD levels (Appendix Fig S3A and B). A marginal increase of CDK2 levels, but not of CDK1 or CDK4, was also observed (Appendix Fig S3A), which could suggest a compensation effect from CDK2 in the absence of CDK8. These data suggest CDK8 may also phosphorylate NICD and regulate its turnover as previously proposed, although in this assay the effect of loss of CDK1 or CDK2 activity upon NICD levels is more pronounced [36,63]. We detected no change in NICD levels following efficient siRNA-mediated depletion of CDK4 in HEK293 cells (Appendix Fig S3C).

Taken together, these data provide further validation of CDK1 and CDK2 involvement in NICD phosphorylation and turnover.

**Pharmacological inhibition of CDK2 and CDK1 activity increases levels of NICD *in vitro***

As a complementary approach to siRNA-mediated loss of function, we selected two small molecule inhibitors that have a highly selective inhibitory activity against CDK2 kinase: Purvalanol B and GSK650394A. The specificity of each of these kinase inhibitors has been tested (at two different concentrations) by the International Centre for Kinase profiling within the MRC Protein Phosphorylation

Unit at the University of Dundee. At 1 and 10 μM, Roscovitine specifically inhibits more than 96% of CDK2 activity. Purvalanol B inhibits more than 95% CDK2 activity, at 0.1 and 1 μM. GSK650394A inhibits 99% of CDK2 activity at both 1 and 10 μM (Fig 4E). CDK2 is by far the most sensitive target for all three inhibitors (Fig 4E). HEK293 cells treated with Roscovitine, Purvalanol B or GSK650394A (10, 0.1 and 10 μM, respectively) for 3 h exhibited significantly increased NICD levels compared to control DMSO-treated cells (Fig 4F and G). iPS cells cultured 3 h in the presence of 0.1 μM Purvalanol B also exhibited significantly elevated levels of NICD as compared to DMSO-treated control cells, indicating this is a conserved effect of CDK2 (Fig EV1E and F).

Given our observation that both CDK1 and CDK2 can phosphorylate NICD peptides *in vitro*, we similarly treated HEK293 cells with a CDK1-specific inhibitor, RO-3306 [68]. Following exposure to 10 μM of RO-3306 for 3 h, HEK293 cells showed elevated NICD levels compared to control DMSO-treated cells (Fig 4H and I). Moreover, Phos-tag technology reveals RO-3306 and Purvalanol B have a reduced number of phospho-bands as compared to MLN4924, indicative of the fact these inhibitors act to reduce NICD phosphorylation and, as seen with MLN4924, the ladder of bands collapses to a single band in the presence of λ phosphatase (Appendix Fig S1B). These data further support the hypothesis that both CDK1 and CDK2 are likely to be involved in phosphorylation-mediated regulation of NICD turnover.

**Levels of NICD increase in HEK293T CDK2$^{−/−}$ cells**

As a complementary strategy to the siRNA-mediated or pharmacological inhibitor methods, we adopted a genetic approach to monitor the effects of loss of CDK2 activity on NICD turnover. To that end, we analysed levels of NICD by Western blot in HEK293T WT and CDK2$^{−/−}$ cells and observed a striking increase in NICD levels in CDK2$^{−/−}$ cells compared to the control (Fig EV2A and B). No changes in CDK1 or CDK8 levels were detected in the CDK2$^{−/−}$ cells. These data validate our previous findings, confirming the implication of CDK2 in regulating NICD turnover. Given that we

have implicated a role for CDK1 in phosphorylating NICD and regulating NICD turnover (Fig 4), we asked whether the simultaneous loss of both CDK1 and CDK2 would lead to an enhanced increase of NICD levels. CDK1 siRNA treatment in WT or $CDK2^{-/-}$ HEK293T cells efficiently depleted CDK1 protein levels, with no effect on levels of CDK2 or CDK8. Under these conditions, NICD levels were significantly increased in $CDK2^{-/-}$ cells compared to control scrambled siRNA-treated $CDK2^{-/-}$ cells. This indicates a synergistic effect following the simultaneous loss of both CDK1 and CDK2 that is greater than that observed through loss of CDK2 alone. CDK8 levels did not change under these conditions (Fig EV2C and D).

Taken together, these data suggest that in the total absence of CDK2, NICD levels are elevated; furthermore, when CDK1 is also depleted in $CDK2^{-/-}$ cells, the increase in levels of NICD is more pronounced. This confirms again that both CDK1 and CDK2 are key kinases in phosphorylating the Notch intracellular domain.

## NICD levels fluctuate over the cell cycle

It is well known that CDK1 and CDK2 share several substrates, with a consequent functional redundancy [69]. Our data demonstrate that these two kinases can phosphorylate NICD in vitro and in the absence of either kinase NICD levels increase in HEK293 cells suggesting they are not acting redundantly in this context. Indeed, loss of both CDK1 and CDK2 activity leads to a synergistic effect upon NICD levels in HEK293T cells. In order to further test this, we analysed whether NICD levels fluctuate during the cell cycle where the role of both CDK1 and CDK2 has been extensively reported in regulating transition to distinct cell cycle phases [70].

To that end, we synchronized HEK293 cells by using a double thymidine block assay. After releasing from the second thymidine block, cells were collected at indicated time points and cell cycle characterization was performed by fluorescence-activated cell sorting (FACS) (Fig 5A).

Figure 5A shows the distribution though the cell cycle of HEK293 cells after synchronization at G1 (0 h), as previously reported [71]. Two and 4 h post-release, the majority of cells were in S phase. At 6 h post-release, the majority of cells were in G2 phase, while 8 h after release, the majority of cells were in late G2/early M phase. At 10 h post-release, the majority of cells had exited mitotic phase and already entered G1. After 12 h from release, cells were in late G1 of the new cell cycle (Fig 5A). The graph in Fig EV3A represents the cell cycle distribution of HEK293 cells at distinct time points post double thymidine block and release from three independent experiments analysed by FACS.

Western blotting analysis of synchronized HEK293 cell extracts showed the expression of several cell cycle regulatory proteins at the same distinct time points reflecting distinct cell cycle phases as described above (Fig 5B). Interestingly, we found that NICD levels fluctuated in a striking manner during the cell cycle, whereas we saw no change to levels of FBXW7. We observed a dramatic decrease of NICD levels at 2 and 4 h corresponding to CDK2-dependent G1/S phase, and again 8 h after double thymidine block release corresponding to the CDK1-dependent G2/M phase transition. These data suggest that NICD levels fluctuate during the cell cycle in a CDK1- and CDK2-dependent manner (Fig EV4). To further explore this, we repeated this Western blot analysis on synchronized HEK293T CDK2 knockout cell extracts. We found a reduced

effect upon fluctuations in NICD levels as compared to control cells (Figs 5B and EV3D). Nevertheless, some fluctuation was observed which may be attributable to functional compensation provided by CDK1 and associated Cyclin activity in the absence of CDK2 [72,73].

In addition, we analysed the cell cycle distribution by flow cytometry in HEK293 cells after CDK2-siRNA-mediated depletion which we have shown leads to a significant increase in NICD levels (Fig 4A) and we observed a slight but nevertheless statistically significant accumulation of cells in G1 phase compared to the control, as expected for cells deprived of CDK2 activity and therefore unable to pass the G1/S checkpoint (Figs 5C and EV3B). These data suggest that the drop in NICD levels occurring in G1/S phase is due to CDK2 phosphorylation of NICD. Similarly, cell cycle distribution, analysed by flow cytometry in HEK293 cells, after CDK1-siRNA-mediated depletion reveals a marginal but nevertheless significant accumulation of cells in G2 phase compared to the control, as expected for cells deprived of CDK1 activity and therefore unable to pass the G2/M checkpoint (Figs 5D and EV3C).

We next investigated whether this fluctuation of NICD levels during different phases of the cell cycle was a conserved phenomenon by quantifying NICD distribution through the cell cycle in a different cell line.

iPS cells were analysed by immunofluorescence using DAPI to mark nuclei and to also distinguish cells in M Phase. The Click-iT EdU assay was used to distinguish cells in S phase, as well as antibodies to NICD and Cyclin A. NICD intensity was measured in order to quantify NICD signal throughout the different stages of the cell cycle. We observed NICD levels were increased significantly during G1 and G2 (Fig 6A and B), which complements our observations in HEK293 cells (Fig 5B). These data demonstrate that fluctuations of NICD levels occur during the cell cycle in a manner that reflects fluctuations in CDK activity (Fig EV4) and this is conserved across different cell types.

## Pharmacological inhibition of CDK2 increases NICD levels and delays the pace of the segmentation clock in the PSM

In order to address whether CDK2 phosphorylation of NICD is involved in driving the NICD-FBXW7 interaction, we performed a co-immunoprecipitation assay with FBXW7 antibody and analysed NICD by Western blot after CDK2 inhibitor treatment. As above, in order to maximize the amount of NICD immunoprecipitated HEK293 cells were treated with MLN4924 (to prevent NICD degradation) +/− Purvalanol B (0.1 μM). A significantly reduced interaction between NICD and FBXW7 was observed after treating HEK293 cells with Purvalanol B for 3 h. This did not reflect a reduction in the level of immunoprecipitated FBXW7 (Fig 7A). Statistical analysis on the density of Western blot bands after immunoprecipitation confirmed an extremely significant reduction in the NICD-FBXW7 interaction following Purvalanol B treatment (Fig 7B). A similar but albeit less pronounced effect was seen with GSK650394A (Appendix Fig S2C and D).

In order to address the potential in vivo role of CDK2-mediated NICD phosphorylation during somitogenesis, we cultured E10.5 mouse PSM explants for 4 h in the presence of 1 μM of Purvalanol B. Initially, we analysed NICD levels by Western blot and just as in the in vitro context, CDK2 inhibition resulted in increased NICD

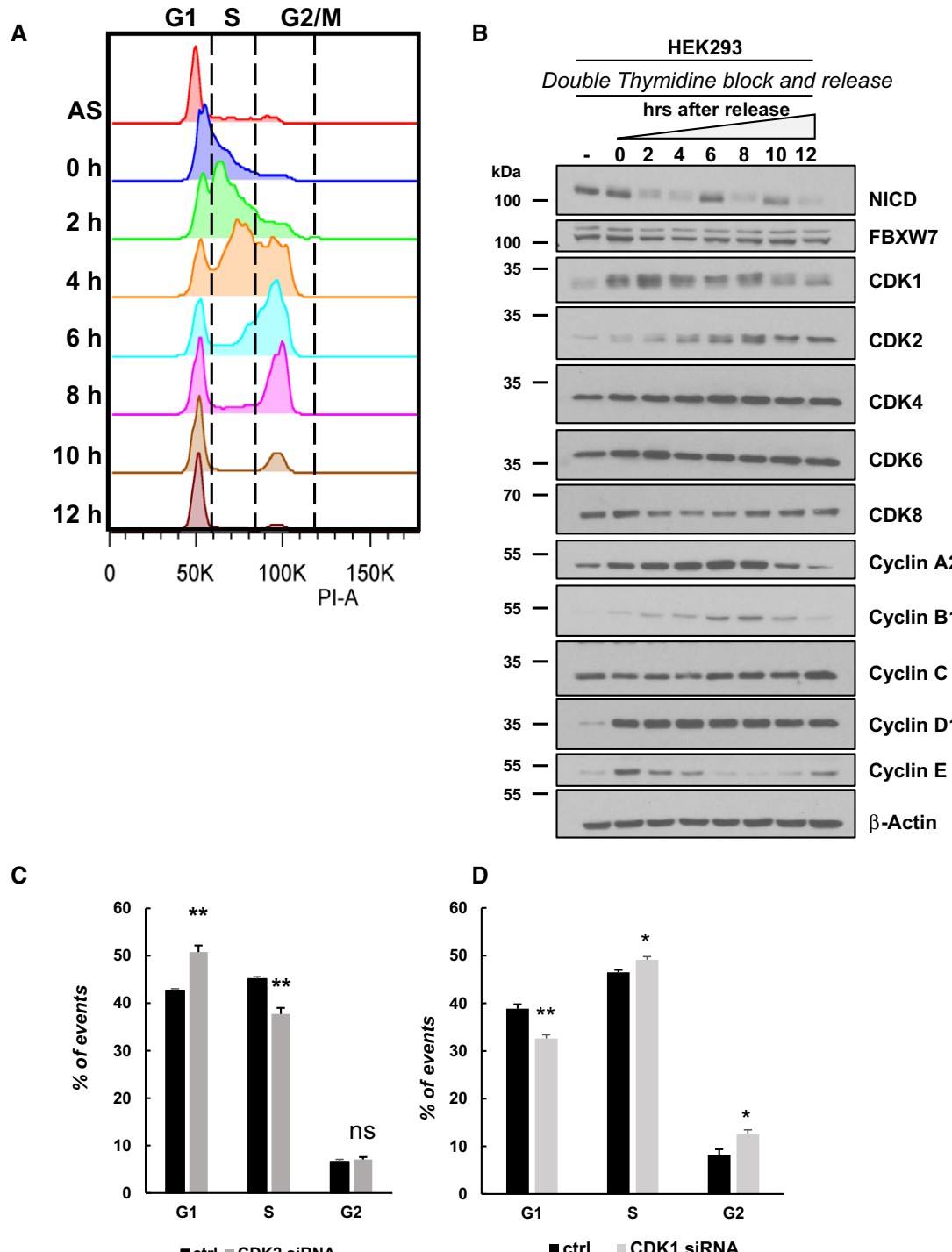

Figure 5. NICD levels fluctuate during the cell cycle.

A   Cell cycle profile for HEK293 cells released from synchronization after double thymidine block. Cells were released and harvested at the indicated time points (AS = asynchronous). Analysis of cell cycle arrest and release was performed using propidium iodide (PI) staining and flow cytometry. A representative experiment of three performed is shown.

B   Expression of the indicated proteins in HEK293 cells was examined by Western blotting, and β-actin was used as loading control. This summary is a representation of three independent experiments.

C, D  Graph of flow cytometry data shows the percentage of cells in given cell-cycle phases 48 h after transfection with plasmids encoding scrambled siRNA or siRNA specific for CDK2 (C) or CDK1 (D). Graphs represent the mean of three independent experiments. All data represent the mean ± SEM from three independent experiments. Student's $t$-test analysis was performed, with *$P \leq 0.05$ and **$P \leq 0.01$ (ns = not significant).

levels as compared to control embryos (Fig 7C and Appendix Fig S3D). This provides the first *in vivo* evidence CDK2 is likely to be involved in NICD turnover.

Previous reports have suggested perturbations to NICD turnover leading to increased NICD levels/stability are closely linked to an increase in the period of segmentation clock oscillations in the PSM [21]. To further explore whether clock gene oscillations were delayed following CDK2 inhibition, we used the half embryo assay, where the PSM from one half of an E10.5 mouse embryo is cultured in control media, while the contralateral half from the same embryo is cultured in the presence of 1 μM of Purvalanol B for 4 h. In 55.5% of cases examined (*n* = 10/18, Table 2A), exposure of PSM explants to Purvalanol B caused a delay in the pace of oscillatory *mLfng* expression across the PSM as compared to the control explant (Fig 7E and Appendix Fig S3Fa-b). Also, in some cases the treated explant ("+") developed one somite less compared to the control (Fig 7E).

Additionally, we cultured E10.5 mouse PSM explants in the presence or absence of the CDK1 inhibitor, RO-3306, for 4 h at 10 μM. Analysis of NICD levels by Western blot revealed that inhibition of CDK1 leads to elevated levels of NICD compared to control embryos (Fig 7D and Appendix Fig S3E). Furthermore, 60% of E10.5 mouse half embryo explants (*n* = 6/10, Table 2B) exposed to RO-3306 treatment for 4 h showed delayed clock oscillations of *mLfng* as compared to DMSO-treated contralateral half embryo explants (Fig 7F and Appendix Fig S3Fc-d).

Finally, considering the assumption that a longer clock period will result in fewer bigger somites in a given time, we performed an analysis similar to that previously reported in Wiedermann *et al* [21] to examine the effect upon somite size following treatment with 1 μM of Purvalanol B or 10 μM of RO-3306 for 7.5 h. By measuring the absolute somite size, we observed that drug-treated embryos display a trend with the last formed somite being larger compared to DMSO control embryos.

In addition, we computed the fold change of successive somite sizes within an embryo (log2 of ratio of somite sizes). Using a nomenclature whereby +1 refers to the most recently formed somite, this corresponds on average to four somites after the drug was added. Thus, by +2 we mean the penultimate somite, which corresponds on average to three somites after the drug was added. Once more, we could appreciate a clear trend for both Purvalanol B and RO-3306-treated embryos showing an increase in the last formed somite, which was not present in control DMSO embryos (Fig 7G and H). These results are consistent with the current understanding that the position of the somite boundaries is already specified for the first few somites that will form from the anterior region of the PSM [74], and thus for those that form just after the time when the drug was added. Our data thus suggest that somite boundary position has been specified at +1, +2, +3 but not at +4.

Taken together, these data not only support our *in vitro* findings, but also importantly provide, for the first time, the *in vivo* evidence that CDK1 and CDK2 are involved in NICD stability and turnover and that this molecular regulation of NICD turnover is extricably linked to the segmentation clock.

### Mathematical model links NICD regulation and cell cycle

To understand how our findings on the molecular details of NICD regulation in individual cells give rise to tissue-scale delay of the segmentation clock, we first developed a mathematical model of NICD production and degradation in HEK293 cells. The variables in the model define the position of a cell in the cell cycle, the amount of NICD and the amount of phosphorylated NICD (pNICD) at time *t*. The model enables us to connect assumptions about molecular processes in individual cells with experiments performed on a large population of cells (see Appendix Fig S4).

By averaging cellular descriptions of NICD and the cell cycle over a given cell population, we derive equations describing the average levels of total NICD (i.e. as measured in a Western blot experiment, see Materials and Methods). Upon release from double thymidine block, cells are synchronized in the cell cycle and there are relatively high levels of CDK activity at different times post-release. As pNICD degrades faster, high levels of CDK results in lower levels of NICD (see Appendix Fig S4 and Figs 4A–D and F–I, and 5B, Appendix Fig S3D and E). We find the model qualitatively fits the experimental observations if NICD is phosphorylated in two separate time windows post-release. We propose that these windows correspond to activity of CDK2 (G1/S phase) and CDK1 (M phase) (Appendix Fig S4A and B). The model can reproduce Western blot results from DMSO and MLN4924 experiments (Appendix Fig S4C). Notably, the simulated Purvalanol B/Roscovitine treatment experiment presented in Appendix Fig S4C is a prediction that is validated by the experimental data (Figs 1B and D, and 4G).

To explore how CDK inhibition results in delay of the segmentation clock, we introduce an additional variable representing phase of the somitogenesis clock and assume that the caudal PSM behaves like a population of phase-coupled oscillators. Given sufficiently strong coupling, a population of such oscillators yields synchronous oscillations with a period that is an average of the individual oscillator periods (Appendix Fig S4E and F). Upon CDK inhibition, the relative number of faster oscillators is reduced; hence, the average period decreases (Appendix Fig S4E). Thus, the cell cycle somitogenesis coupled model provides a description of how CDK-mediated phosphorylation of NICD can result in the observed phenotype in PSM tissue (Fig 7).

## Discussion

In this study, we report for the first time that cell cycle-dependent CDK1 and CDK2 activity is involved in NICD turnover, linking NICD turnover with the cell cycle which has broad implications across all developmental and disease contexts where Notch plays a role.

We demonstrate that inhibitors (Roscovitine, DRB and XAV939), previously shown to prolong NICD half-life and delay the segmentation clock pace in mouse and chick PSM *in vivo* [21], also increase endogenous levels of human NICD when a range of primary human cell lines were treated for 3 h. This highlights the conserved effect of these inhibitors on regulating NICD stability, which is perhaps not surprising given the high degree of sequence similarity between mouse, human and chicken NICD. While it is true we cannot formally exclude the possibility that the inhibitors may also affect full length Notch, it is nevertheless clear they have an effect on NICD stability since in the presence of LY411575, which inhibits Notch processing, we see an increase in levels of the processed form.

A number of reports have highlighted the fact that the SCF[FBXW7] E3 ligase plays an important role in NICD degradation

A

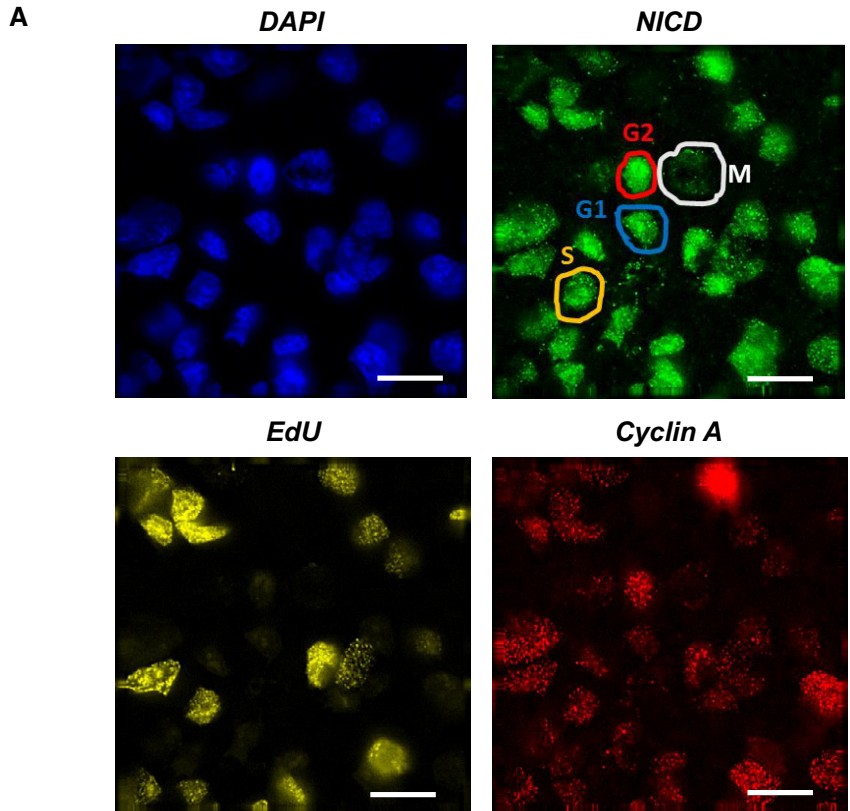

B

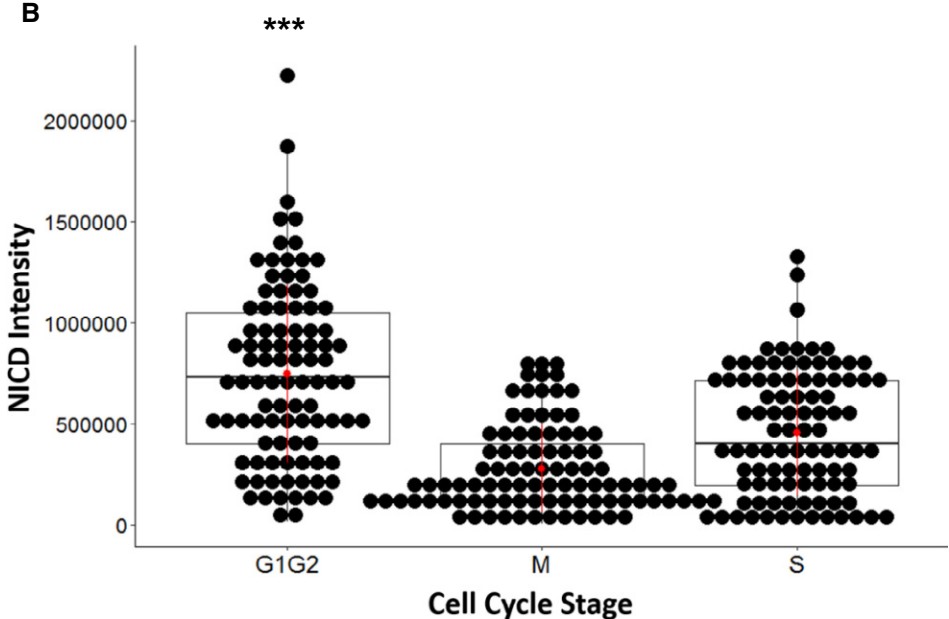

**Figure 6. NICD levels peak in G1 and G2 phases of the cell cycle for iPS cells.**

A   iPS cells in G2 express high levels of Cyclin A and do not incorporate EdU (marked in red within the NICD panel). EdU incorporation occurs during S phase (marked by yellow in the NICD panel). Cells with low Cyclin A intensity and no EdU incorporation are either in G1 or M phase. M phase cells are distinguishable by their chromatin condensation, observable in the DAPI panel (dividing cells are indicated with white in the NICD panel). G1 cells do not have any of the features described above (blue in NICD panel). Scale bars are 20 μm.

B   G1 and G2 phase cells have significantly higher NICD protein levels than S and M stage cells ($P \leq 0.001$). All groups were compared against each other with a one-way ANOVA analysis, followed by Tukey's test. M phase cells have the lowest NICD. Each point in the box-plot represents the normalized intensity of a single cell. Data from three different experiments ($n = 3$) were randomized collated and plotted.

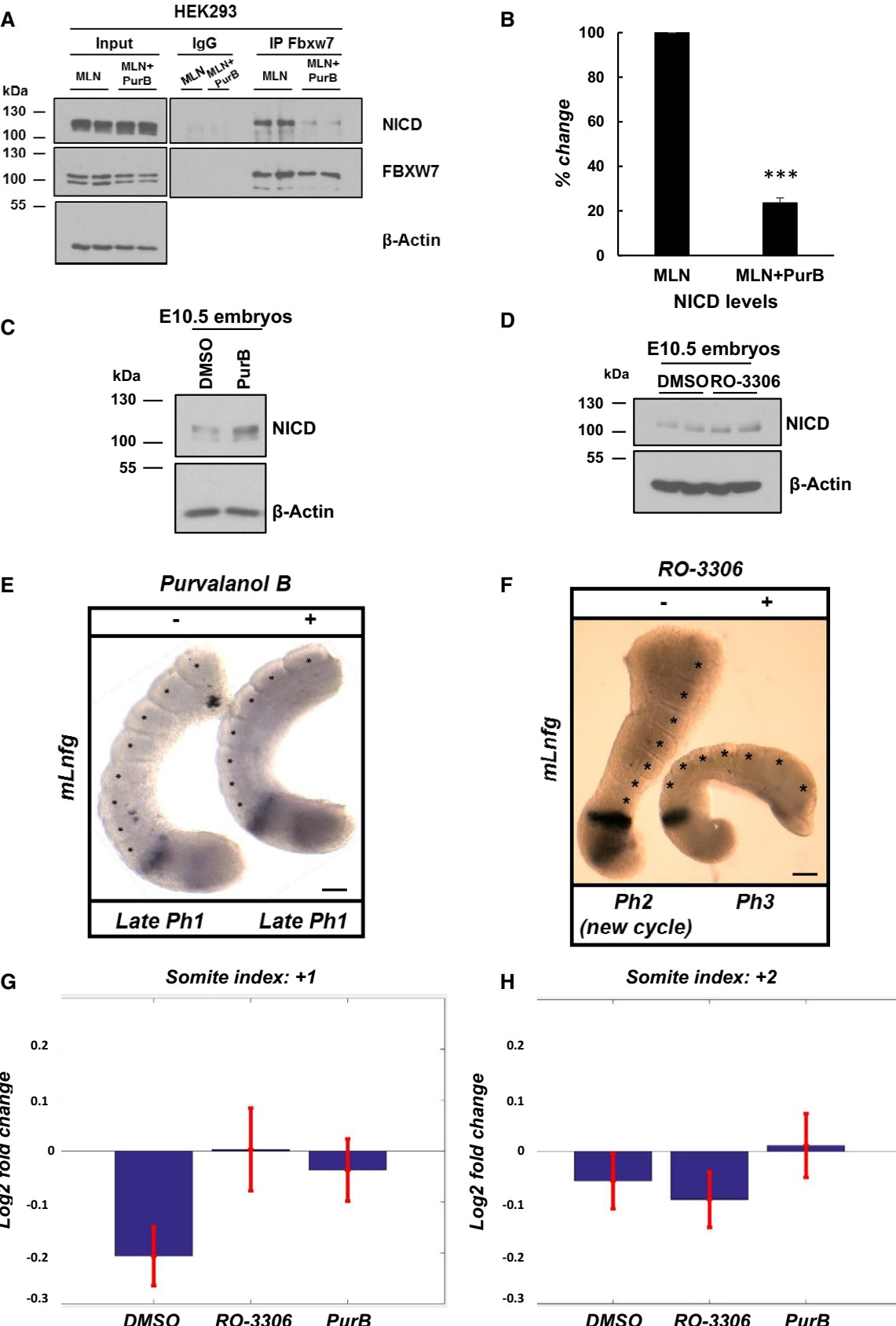

Figure 7.

◄

**Figure 7.  Purvalanol B treatment reduces the NICD-FBXW7 interaction and CDK1 or CDK1 inhibition delays the pace of the segmentation clock.**

A   Purvalanol B treatment reduced the NICD-FBXW7 interaction. 200 µg of HEK293 cell lysates treated with MLN4924 or MLN4924 in combination with 0.1 µM of Purvalanol B was subjected to immunoprecipitation using FBXW7 antibody, or IgG antibody as negative control, and precipitated material was analysed by Western blot using NICD antibody. Western blot with FBXW7 antibody served as loading control for immunoprecipitation efficiency. 10% of cell lysate before immunoprecipitation was used as input control and β-actin has been used as loading control.

B   Quantification of the density of Western blot bands in (A) performed by ImageJ software. Data are expressed as percentage changes compared to MLN4924-treated samples. All data represent the mean ± SEM from three independent experiments. Student's *t*-test analysis was performed, with ***$P \leq 0.001$.

C   E10.5 mouse tails were bisected down the midline. One half (+) was cultured for 4 h in the presence of Purvalanol B (1 µM). The contralateral half (−) was cultured for 4 h in the presence of DMSO. Control or treated explants were pooled, and NICD levels were detected by Western blot. β-Actin was used as loading control.

D   E10.5 mouse tails were bisected down the midline. One half (+) was cultured for 4 h in the presence of RO-3306 (10 µM). The contralateral half (−) was cultured for 4 h in the presence of DMSO. Control or treated explants were pooled, and NICD levels were detected by Western blot. β-Actin has been used as loading control.

E   Bisected E10.5 mouse PSM explants were cultured in the absence (−) or presence (+) of 1 µM of Purvalanol B for 4 h and then analysed by *in situ* hybridization for *mLfng* mRNA expression. Purvalanol B-treated explant has one less somite than the control explant, and the treated explant is in the same late phase 1 of the oscillation cycle of dynamic *mLfng* mRNA expression indicating it is a whole cycle delayed compared to the "−" explant. *n* = 18. Scale bar is 100 µm.

F   Bisected E10.5 mouse PSM explants were cultured in the absence (−) or presence (+) of 10 µM of RO-3306 for 4 h and then analysed by *in situ* hybridization for *mLfng* mRNA expression. RO-3306-treated explant is two phases behind in the oscillation cycle of dynamic *mLfng* mRNA expression indicating there is a delay in the oscillation compared to the "-" explant. *n* = 10. Scale bar is 100 µm.

G   The mean log2 fold change in successive somite length at position +1 (last formed somite) is plotted for different treatment conditions. Error bars denote standard error of the mean. Paired *t*-test analysis was performed at 5% significance level, with *P* = 0.0437 for RO-3006 and *P* = 0.0524 for PurB, compared to DMSO. *n* = 14.

H   The mean log2 fold change in successive somite length at position +2 (penultimate formed somite) is plotted for different treatment conditions. Error bars denote standard error of the mean. Paired *t*-test analysis was performed at 5% significance level, with *P* = 0.6409 for RO-3006 and *P* = 0.4065 for PurB, compared to DMSO. *n* = 14.

[36,38,46–48,64]. To date, the interaction between NICD and FBXW7 by co-immunoprecipitation has only been shown using overexpressed proteins, due to the fact this interaction is very transient and leads to efficient degradation of NICD [36,38,47,48,64]. We demonstrate for the first time that NICD and FBXW7 can interact at endogenous levels in HEK293 cells. Moreover, this allowed us to demonstrate that, when phosphorylation is disrupted by small molecule CDK inhibitors (Roscovitine, Purvalanol B or DRB), the NICD-FBXW7 interaction is reduced. It is important to note that none of these inhibitors abolished the NICD-FBXW7 interaction, which suggests again that they are each inhibiting only some of the kinase activity involved in NICD phosphorylation and subsequent recruitment of FBXW7. This aligns with the observation that in the chicken/mouse PSM both of these inhibitors increase NICD stability and the period of the segmentation clock but that NICD turnover still occurs in this tissue and thus is dependent on a number of different kinases/phosphorylation events differentially targeted by these two inhibitors [21].

By mass spectrometry analysis, we identified 15 phospho-sites within human exogenous NICD in HEK293 cells, some of which have been previously identified and reported to be involved with NICD turnover [36,63]. Among those phospho-sites identified, we found that Serine 2513, when mutated to alanine, thereby rendering the site non-phosphorylatable, was essential for the interaction between NICD and FBXW7. Point mutations in a number of other phosphorylated residues showed these are non-essential for this NICD-FBXW7 interaction, including two residues that have previously been reported to potentially be required for the interaction; S2484 and S2506 (S2516 and S2538 in our annotation) [36,64]. However, Fryer and colleagues used exogenous proteins and mutated all three sites simultaneously, so the individual contribution/requirement of each residue was not addressed in that study. O'Neil and colleagues have also reported, using exogenous proteins, that phosphorylation of the threonine residue at T2487 in mouse NICD (T2511 in our annotation) is key to driving the NICD-FBXW7 interaction and subsequent ubiquitination of NICD [38]. We did not observe phosphorylation at this residue in human NICD. However,

it would be interesting to repeat the mass spectrometry analysis using different enzyme digestions to determine whether this reveals additional phosphorylated sites that play a role in this interaction.

We have identified CDK1 and CDK2 as two kinases that can phosphorylate NICD in the PEST region that harbours residue serine 2513, which we have demonstrated to be crucial for the interaction between NICD and FBXW7. Through three loss-of-function approaches, namely siRNA, generation of CDK2 knockout cell line and a pharmacological approach, we further demonstrated that inhibiting phosphorylation by CDK1 or CDK2 renders NICD more stable.

CDK1 and CDK2 activity, and thus phosphorylation of their substrates, changes in a cell cycle-dependent manner [75]. Strikingly, we find that NICD levels vary in HEK293 cells and iPS cells in a cell cycle-dependent way such that we observe lowest levels of NICD in phases of the cell cycle where CDK1/Cyclin B1 and CDK2/Cyclin E levels are highest and therefore when these complexes are reported to be most active [76]. This finding suggests that NICD activity, signal duration and signal strength are likely to vary in a cell cycle-dependent manner, potentially leading to differential transcriptional outputs in different phases of the cell cycle.

Given our findings, it is striking that CDK2 homozygous *null* mice are viable [72]. However, it has been reported that CDK1 and CDK2 share more than 50% of their targets [77] which is likely to allow for some redundancy in this knockout line, as CDK1 can compensate for loss of CDK2 by forming active complexes with A-, B-, E- and D-type Cyclins. Indeed, we find inhibiting both CDK1 and CDK2 activity simultaneously in HEK293T cells led to a synergistic effect on NICD levels. The CDK1-*null* homozygous mice however are embryonic lethal, and a CDK2 knock-in to the CDK1 locus is unable to rescue this phenotype [78,79]. It would be interesting to examine the role of the CDK1 conditional mutant in the context of Notch signalling and somitogenesis [80]. Interestingly, Cyclin C-*null* mice, which are embryonic lethal at 10.5, show dramatically reduced NICD1 phosphorylation *in vivo* and elevated NICD1 levels. The authors show that Cyclin C can complex with CDK3, CDK19, CDK8, CDK1 and CDK2 to phosphorylate NICD1 and promote

**Table 2.** Purvalanol B and RO-3306 small molecule inhibitors delays the pace of the segmentation clock.

| Sample no. | Before treatment | DMSO | | PurB | |
|---|---|---|---|---|---|
| | No. of somites | No. of somites | Phase | No. of somites | Phase |
| (A) | | | | | |
| 1 | 6.5 | 8 | Ph2 | 7 | Ph3 |
| 2 | 4.5 | 6.5 | Late Ph1 | 6.5 | Ph3 |
| 3 | 4.5 | 7.5 | Ph3 | 6.5 | Ph2 |
| 4 | 6 | 8 | Early Ph2 | 8 | Early Ph2 |
| 5 | 8.5 | 11 | Ph3 | 11 | Ph3 |
| 6 | 6 | 9 | Ph3 | 9 | Ph3 |
| 7 | 7 | 9 | Early Ph2 | 9 | Early Ph2 |
| 8 | 6 | 7 | Ph3 | 7 | Ph3 |
| 9 | 6 | 7 | Late Ph3 | 6 | Late Ph2/early Ph3 |
| 10 | 6 | 7 | Ph2 (new cycle) | 7 | Ph3 (previous cycle) |
| 11 | 6 | 7 | Ph3 | 7 | Ph3 |
| 12 | 4.5 | 5.5 | Ph3 | 5.5 | Ph3 |
| 13 | 4 | 5 | Ph3 | 5 | Ph3 |
| 14 | 6 | 7 | Ph3 | 6 | Ph2 |
| 15 | 6 | 7 | Ph3 | 6 | Ph3 |
| 16 | 5.5 | 8 | Late Ph1 | 7 | Late Ph1 |
| 17 | 7 | 9 | Ph2 | 8 | Ph2 |
| 18 | 5 | 6 | Ph2 | 5 | Ph2 |

| Sample no. | Before treatment | DMSO | | RO-3306 | |
|---|---|---|---|---|---|
| | No. of somites | No. of somites | Phase | No. of somites | Phase |
| (B) | | | | | |
| 1 | 6 | 8 | Ph3 | 7 | Ph2 |
| 2 | 3.5 | 5 | Late Ph3 | 4 | Late Ph2/early Ph3 |
| 3 | 5.5 | 7 | Ph2 (new cycle) | 7 | Ph3 |
| 4 | 5 | 6 | Ph3 | 6 | Ph2 |
| 5 | 3.5 | 6 | Ph3 | 4 | Ph2 |
| 6 | 4 | 6 | Ph3 | 5 | Ph3 |
| 7 | 5 | 7 | Ph3 | 7 | Ph3 |
| 8 | 4 | 6 | Ph3 | 6 | Ph3 |
| 9 | 4 | 6 | Early Ph3 | 5 | Ph2 |
| 10 | 4.5 | 6 | Ph3 | 6 | Ph3 |

Summary table of number of somites and clock delays before and after treatment with Purvalanol B (A) and RO-3306 (B) inhibitors, compared to DMSO, in E10.5 mouse explants.

NICD1 degradation [63]. Thus, it would be very interesting to examine a PSM conditional Cyclin C loss of function to determine whether this Cyclin is involved in regulating NICD turnover in this tissue. It is noteworthy Cyclin C levels did not appear to vary in a cell cycle-dependent manner in HEK293 cells, although it is possible the activity of Cyclin C/CDK complexes varies in a cell cycle-dependent manner through post-translational modifications rather than protein levels *per se*.

The finding that CDK1 or CDK2 inhibition leads to increased levels of NICD in E10.5 mouse embryo PSM lysates and to a delay

in clock gene oscillations and somite formation provides the first *in vivo* mechanistic evidence that both CDK1 and CDK2 are involved in NICD turnover.

It is noteworthy that inhibition of CDK2 activity with a highly selective inhibitor reduces the NICD-FBXW7 interaction in HEK293 cells but does not block it completely. This suggests that some NICD turnover persists under these conditions, possibly through redundancy between CDK1 and CDK2 and/or through CDK8-mediated phosphorylation, the subsequent recruitment of E3 ligases and degradation of NICD. Moreover, a report by Chiang *et al* has also

identified a region downstream of the PEST sequence, termed S4, that is involved in NICD degradation, but that is independent of FBXW7 activity [81]. These data indicate there are several mechanisms regulating NICD turnover, which are partially redundant.

We also developed a mathematical model that coupled cell cycle dynamics to NICD degradation. Using HEK293 cells, model parameters were identified that recapitulated the distributions of cells in the different cell cycle phases. To recover qualitative features of NICD thymidine release experiments, NICD phosphorylation was required at distinct stages in the cell cycle. After using MLN4924 and control experiments to estimate NICD phosphorylation rate, the model qualitatively predicts the effect of Roscovitine/PurB treatment on total NICD levels. To address the likely effect of CDK2 inhibition in the PSM, the model was extended to account for position of a cell in the segmentation clock cycle. Following Wiedermann *et al* [21], it was assumed that levels of NICD are anti-correlated with clock somitogenesis frequency. Hence, the posterior PSM is represented by a population of phase-coupled oscillators whose frequency is cell cycle dependent. Simulating CDK2 inhibition removes a pool of faster oscillators thus reducing the tissue period.

Notch plays a key role as a gatekeeper protecting progenitor and/or stem cells in multiple developmental contexts, in part through preventing differentiation and in part through regulating components of the cell cycle [82–84]. Our novel finding of a reciprocal auto-regulatory role between the cell cycle regulated CDKs, CDK1 and CDK2, and NICD turnover has potentially great relevance to the developmental biology community and may provide additional insight into disease/cancer contexts where this autoregulation may have gone awry.

# Materials and Methods

All plasmids and reagents indicated as such in the text are available from MRC-PPU Reagents and Services (https://mrcppureagents.dundee.ac.uk/).

## Cell culture

HEK293 (human embryo kidney cells) and IMR90 (Human Caucasian fetal lung fibroblast) cells were obtained from the American Type Culture Collection (ATCC). Cells were routinely cultured and maintained in DMEM High glucose (Gibco) supplemented with 10% Fetal Bovine Serum (FBS; LabTech), 1% Penicillin/Streptomycin (Gibco), 1% Sodium Pyruvate (Gibco) and 2 mM L-Glutamine (Gibco).

mESCs (mouse embryonic stem cells) were cultured on gelatin-coated plates in media containing LIF, 10% fetal calf serum (Gibco) and 5% knockout serum replacement (Invitrogen).

ChiPS4 human-induced pluripotent stem (iPS) cells (derived from new born human dermal fibroblasts) were purchased from Cellartis AB and maintained using DEF-CS (Cellartis AB) according to the manufacturer's recommendations. For experiments, cells were seeded as single cells on Geltrex™-coated dishes (10 μg/cm²) in DEF medium supplemented with 10 μM of Rho-kinase inhibitor Y27632 (Tocris) at a density of $1 \times 54$ cells/cm² and allowed to attach overnight. The medium was then replaced with fresh DEF medium (Y27632), and after a further 24 h, cells were treated with inhibitors.

The CDK2 homozygous mutant cell line was generated in HEK293T cells (ATCC CRL-3216, Manassas, VA). The Integrated DNA Technologies (IDT, Coralville, IA) Alt-R CRISPR-Cas9 ribonucleoprotein system was used.

Briefly, a custom crRNA to the human CDK2 locus was designed using the optimized primer design at crispr.mit.edu (CAGAAACAA GUUGACGGGAG-GUUUUAGAGCUAUGCU). Cas9:crRNA:tracrRNA, ATTO™-550 ribonucleoprotein complexes were transfected into 293T cells with Lipofectamine RNAiMAX transfection reagent (#13778100, Invitrogen, Grand Island, NY) following the recommended protocol from IDT for cationic lipid delivery into mammalian cells. The fluorescent tag labelled tracrRNA, ATTO™-550, was used to sort for transfected cells 24 h post-transfection. ATTOTM550-positive cells were single cell sorted into 96-well plates and expanded. Clones were screened for deletions by nested PCR (Forward primer: CAAATTGACAAGAGCGAGAGG, Internal primer: AAGTTGACGGGAGAGGTGGT, Reverse primer: AGTCAGGGTGTG GTTTTTGG) and confirmed by Sanger sequencing.

All cells were grown at 37°C in 5% $CO_2$. Cells were trypsinized and split into new plates at subconfluency.

## Transfections

### Overexpression transfections

HEK293 cells were seeded 24 h prior to transfection and then transiently transfected following the GeneJuice® Transfection Reagent protocol provided by Merck Millipore. Depending on plate size, 2–9 μg of plasmid was used. Cells were harvested 24 h after transfection.

### siRNA transfections

Small interfering RNA oligonucleotides were purchased from MWG/Eurofins and used in a stock concentration of 20 μM. siRNAs were transfected using calcium phosphate transfection method [85]. siRNA oligonucleotides were diluted in a solution containing distilled sterile $H_2O$ and 2M $CaCl_2$.

The mixture was then added dropwise to a round-bottom 15-ml tube containing 2× HBS (0.137M NaCl, 0.75M Na2HPO4, 20 mM HEPES pH 7.0). The content was then added to cells and incubated overnight at 37°C.

Media was changed after 24 h, and cells were harvested after 48 h of transfection.

## Oligonucleotide sequences for siRNA knockdown siRNA (5′→3′)

*Control:* AACAGUCGCGUUUGCGACUGG; *CDK1:* AAGGGGUUCC UAGUACUGCAA; *CDK2:* CCUCAGAAUCUGCUUAUUA; *CDK4:* AA GCCGACCAGUUGGGCAAAA; *FBXW7:* ACAGGACAGUGUUUACAAA. Commercial CDK8 siRNA (Cell Signalling, #6438) has been used.

## Plasmids and mutagenesis

hNICD-GFP vector was generated and obtained from MRC-PPU reagents, University of Dundee. All NOTCH1 mutants were generated and obtained from MRC-PPU reagents, University of Dundee.

Briefly, the fragment NOTCH1 1,754–2,555 (end) was synthesized by GeneArt with flanking BamHI and NotI restriction sites to facilitate cloning, and the sequence was codon-optimized for

mammalian expression. This was then digested and ligated into expression vector pCMV5D GFP to make the wild-type clone pCMV5D GFP NOTCH1 1,754-end.

Site-directed mutagenesis was carried out using the QuikChange Lightning Site-Directed Mutagenesis Kit (Agilent Technologies) but substituting the Taq with KOD Hot Start DNA polymerase (Novagen). All mutations were confirmed by sequencing.

S2205A change was introduced using primers
Forward: 5′-CCCGTGGATAGCCTGGAAGCGCCTCACGGCTACCTGAGC
Reverse: 5′-GCTCAGGTAGCCGTGAGGCGCTTCCAGGCTATCCACGGG

S2513A change was introduced using primers
Forward: 5′-CCCATTTCTGACCCCTGCACCCGAGAGCCCCGATC
Reverse: 5′-ATCGGGGCTCTCGGGTGCAGGGGTCAGAAATGGG

S2516A change was introduced using primers
Forward: 5′-CTGACCCCTAGCCCCGAGGCGCCCGATCAGTGGTCTAGC
Reverse: 5′-GCTAGACCACTGATCGGGCGCCTCGGGGCTAGGGGTCAG

S2513A and S2516A double mutant was generated with primers
Forward: 5′-CACCCCTTCCTCACCCCGGCCCCTGAGGCCCCTGAC
CAGTGGTCCA
Reverse: 5′-TGGACCACTGGTCAGGGGCCTCAGGGGCCGGGGTGAG
GAAGGGGTG

S2527A change was introduced using primers
Forward: 5′-TCTAGCAGCAGCCCCCACGCGAACGTGTCCGATTGGAGC
Reverse: 5′-GCTCCAATCGGACACGTTCGCGTGGGGGCTGCTGCTAGA

S2538A change was introduced using primers
Forward: 5′-TGGAGCGAAGGCGTGTCCGCCCCCCCAACCAGCATGCAG
Reverse: 5′-CTGCATGCTGGTTGGGGGGGCGGACACGCCTTCGCTCCA

T2511V mutant was generated with primers
Forward: 5′-CCCGAGCACCCATTTCTGGTCCCTAGCCCCGAGAGCCCC
Reverse: 5′-GGGGCTCTCGGGGCTAGGGACCAGAAATGGGTGCTCGGG

P2512L change was introduced using primers
Forward: 5′-GAGCACCCATTTCTGACCCTTAGCCCCGAGAGCCCCGAT
Reverse: 5′-ATCGGGGCTCTCGGGGCTAAGGGTCAGAAATGGGTGCTC

P2514R change was introduced using primers
Forward: 5′-CCATTTCTGACCCCTAGCCGCGAGAGCCCCGATCAGTGG
Reverse: 5′-CCACTGATCGGGGCTCTCGCGGCTAGGGGTCAGAAATGG

## Treatments

Cells were treated for 3 h with different drugs: 150 nM of LY411575 (generated in house, University of Dundee) [9], 1 μM of MLN4924 (MRC-PPU reagents, University of Dundee), 10 μM of 5,6-Dichloro-1-beta-D-ribofuranosylbenzimidazole (DRB) (Sigma), 10 μM of Roscovitine (Calbiochem), 10 μM of XAV939 (Tocris Bioscience), 0.1 μM of Purvalanol B (MRC-PPU reagents, University of Dundee), 10 μM of GSK650394A (MRC-PPU reagents, University of Dundee), 10 μM of RO-3306 (Sigma) and DMSO (Sigma) as control.

Lysates were treated with λ-phosphatase (New England BioLabs) following the manufacturer's instructions.

iPS cells were incubated with nucleotide analogue EdU (Invitrogen) with a final concentration of at 40 μM.

## Protein extraction

Cells were lysed on ice for 15 min by adding the appropriate volume of lysis buffer (50 mM Tris–HCl pH 7.5, 150 mM NaCl, 1 mM EDTA, 1 mM EGTA, 10 mM NaF, 1 mM Na3VO4, 0.1% β-mercaptoEtOH, 1 tablet of protease inhibitor cocktail, Thermo Scientific).

Cells were harvested by scraping, and the lysate was clarified by centrifugation for 15 min at 14,000 *g* at 4°C. The supernatants were collected and stored at −80°C.

Bradford reagent (Bio-Rad) has been used as the colorimetric assay for protein measurements.

## Western blot

20 to 50 μg of protein was diluted 1:1 with 2× loading buffer (100 mM Tris–HCL pH 6.8, 20% glycerol, 4% SDS, 200 mM DTT and Bromophenol Blue), and proteins were denatured for 5 min at 95°C. Samples were resolved on a SDS–PAGE following standard procedures.

Gels were transferred onto Nitrocellulose membrane (GE Healthcare) for 1.5 h at 400 mA. The membrane was then blocked with 5% Milk in TBS-tween buffer (20 mM Tris pH 7.6, 150 mM NaCl, 0.1% Tween) for 20 min. Membranes were incubated 30 min/1 h or O/N at 4°C in primary antibodies [Cleaved Notch1 (1:1,000, Cell Signalling #4147); β-Actin (1:10,000, Proteintech #66009-1-Ig); Fbxw7 (1:1,000; Abcam #171961); GFP (1:1,000, Cell Signalling #2956); Cdk1 (1:1,000, Cell Signalling #9116); Cdk2 (1:1,000, Cell Signalling #2546); Cdk4 (1:1,000, Santa Cruz #260); Cdk6 (1:1,000, Santa Cruz #177); Cdk8 (1:1,000, Bethyl #A302-501A-T); Cyclin A2 (1:1,000, Cell Signalling #4656); Cyclin B1 (1:1,000, Cell Signalling #4138); Cyclin C (1:1,000, Bethyl #A301-989A-M); Cyclin D1 (1:5,000, Abcam #137875); Cyclin E1 (1:1,000, Cell Signalling #4129); Total histone H3 (1:1,000, Cell Signalling #9715); Phospho-histone H3 (1:1,000, Cell Signalling #3377)].

The membranes were then washed with TBS-Tween and incubated with the appropriate secondary HRP antibody (Cell Signalling, #7074, #7076). After washing, membranes were developed using ECL solution (Pierce).

## Western blot quantification

For band intensity quantification method, exposed and developed films were scanned in CanoScan LiDE60 scanner.

Band intensity was quantified by ImageJ software. The obtained images after scan were converted to 8-bit format in order to perform uncalibrated optical density (OD).

Bands were selected and circumscribed with the rectangular ROI selection and "Gels" function, followed by quantification of peak area of obtained histograms. Data were acquired as arbitrary area values.

To ensure band intensity accurately reflects protein abundance, this has been compared to one invariant normalization controls, such as actin, as housekeeping proteins.

Furthermore, the linear range of detection of protein concentration, which can be used for reliable quantitation, has been determined by performing a dilution series of a representative experimental sample (Fig EV1D).

## Immunoprecipitation

200 μg to 1 mg of proteins was incubated O/N at 4°C on rotation with 2 μg of Fbxw7 antibody (Bethyl, A301-721A). Rabbit IgG was used as negative control. Protein G Agarose beads (Cell Signalling #37478) were added for 2 h at 4°C on rotation. Precipitates were washed three times with cold PBS before adding 2× loading buffer.

Samples were denatured for 5 min at 95°C and then analysed by Western blot.

Immunoprecipitation using GFP-Trap®_A beads (ChromoTek) was performed according to the manufacturer's instructions.

### Phos-tag gel

Prior loading into the gels for Phos-tag SDS–PAGE, samples were supplemented with 10 mM MnCl$_2$. Phos-tag SDS–PAGE was carried out as previously described [62,86]. After electrophoresis, gels were washed three times for 10 min each in the transfer buffer [48 mM Tris/HCl, 39 mM glycine and 20% (v/v) methanol] containing 10 mM EDTA and 0.05% (w/v) SDS, followed by one wash in the transfer buffer containing 0.05% SDS for 10 min. Proteins were transferred and incubated with specific antibodies as previously described.

### Mass spectrometry

Upon separation by SDS–PAGE on NuPAGE™ 4-12% Bis-Tris protein gels (Thermo Scientific) and staining with Instant Blue (Expedeon), protein bands were excised from gels. Samples were reduced with 10 mM DTT at 50°C for 30 min and alkylated with 100 mM IAA at room temperature for 20 min in the dark.

Digestion was carried out by adding sequencing grade chymotrypsin (Sigma) at a ratio of 1–50 (enzyme to substrate) and incubating O/N at 30°C.

Peptides were extracted with 100% ACN containing 2.5% formic acid and dried in a vacuum centrifuge.

Mass spectrometric analysis was performed by LC-MS-MS using a linear ion trap-orbitrap hybrid mass spectrometer (LTQ-Orbitrap Velos, Thermo Fisher Scientific) and coupled to a Dionex Ultimate 3000 nanoLC system. Peptides were typically injected onto a Thermo Scientific 15 cm Easy spray column (Part No. ES800), with a flow of 300 nl/min and eluted over a 40 min linear gradient of 97% solvent A (3% DMSO, 2% Acetonitrile, 0.1% formic acid in H$_2$O) to 35% solvent B (90% acetonitrile, 3% DMSO, 0.08% formic acid in H$_2$O). Data files were analysed by Proteome Discoverer 2.0 (Thermo), using Mascot 2.4.1 (www.matrixscience.com), and searching against the Swissprot and MRC-PPU (University of Dundee) databases. Scaffold (www.ProteomeSoftware.com) was also used to examine the Mascot result files. Allowance was made for the following modifications: fixed, Carbamidomethyl (C), and variable, Oxidation (M), Dioxidation (M) and Phosphorylation (S, T and Y). Error tolerances were 10 ppm for MS1 and 0.6 Da for MS2. Phospho-sites were assigned according to Proteome Discoverer ptmRS when the phospho-RS site probability was ≥ 90%.

The mass spectrometry proteomics data have been deposited to the ProteomeXchange Consortium via the PRIDE [87] partner repository with the dataset identifier PXD013038 and 10.6019/PXD013038.

### Kinase assay

Kinase assay was designed and performed by MRC-PPU International centre for Kinase Profiling at University of Dundee (http://www.kinase-screen.mrc.ac.uk/). All assays were carried out using a radioactive (33P-ATP) filter-binding assay.

### Fluorescence-activated cell sorter analysis

HEK293 cells were harvested, pelleted by gentle centrifugation (450 $g$ × 5 min) and washed once in 1 ml of PBS + 1% v/v FBS and transferred to FACS tubes (Scientific Laboratory Supplies).

Cells were pelleted again and then fixed in cold 70% ethanol for 30 min at room temperature.

Cells number was adjusted to approximately 5 × 10$^5$ cells, and then, cells were washed twice in PBS +1% v/v FBS.

Cell pellet was then resuspended in 300 μl of staining buffer (50 μg/ml propidium iodide, 50 μg/ml RNase A in PBS +1% v/v FBS) and incubated for 30 min in the dark at room temperature.

FACS analyses were conducted by flow cytometry and cell sorting facility (University of Dundee).

Briefly, samples were analysed on a FACS Canto II flow cytometer from Becton Dickinson. Propidium iodine was detected using 488 nm excitation and emission collected at 587 ± 40 nm. Data were analysed using Flowjo software (Flowjo, LLC). Single cells were identified on the basis of PI-A and PI-W measurements and doublets excluded. Cell cycle distribution of the resulting PI-A histograms was determined using the Watson-Pragmatic model.

### Cell cycle synchronization

To synchronize HEK293 or HEK293T CDK2 KO cells at G1/S, a double thymidine block assay was performed. Cells were treated with 2.5 mM of thymidine (Sigma) for 18 h, washed twice with PBS, released into fresh media for 9 h, treated for a further 14 h with 2.5 mM of thymidine and then released again into fresh media following two washes in PBS. Cells were then collected for FACS and Western blot protein analyses at the indicated time points, following protocols already described.

### Immunofluorescence microscopy and image analysis

iPS cells (ChiPS4, Cellartis) were washed with PBS and fixed with 2% v/v paraformaldehyde (PFA) at room temperature for 20 min and washed with PBST. Cells were permeabilized with 0.5% Triton X-100 and 0.5% Tween-20 in PBS for 10 min and washed twice for 5 min with PBST and then for 5 min with PBS. Antigen retrieval was performed using 1% v/v sodium dodecyl sulphate (SDS) for 7 min, followed by a 10-min incubation in PBS and 5-min PBST washes. Coverslips were then blocked in blocking solution (0.5% fish scale gelatine (FSG), 0.5% Triton X-100 and 0.5% Tween-20 in PBS) for 90 min. Cells were incubated with Cleaved Notch 1 (Cell Signalling #4147; 1:100) and Cyclin A (Santa Cruz #271645; 1:200) primary antibodies. Control coverslips were left in the blocking solution for the duration of the primary incubation. Following primary antibody, cells were washed four times for 5 min each in PBST and afterwards incubated in secondary AF488 Donkey anti-rabbit (Abcam #21206) and AF647 Donkey anti-mouse (Invitrogen #A31571) antibodies diluted 1:200 in block solution for 1 h at RT, in dark. Cells were washed with PBST and then with PBS. To visualize nucleotide incorporation during S phase, coverslips were incubated in Click-iT EdU reaction mix with Azide 555 (Thermo Fisher Scientific #C10338), following manufacturer's protocol, for 30 min. Cells were then stained with DAPI (1:1,000 from 1 mg/ml stock) for

10 min at RT and mounted on Superfrost Plus microscope slides (Thermo Scientific) with VectaShield.

Images were acquired using the Delta Vision "Mercury" deconvolution microscope. An Olympus 60× oil immersion objective was used, and images were captured using a CCD camera. Images were deconvoluted and analysed using ImageJ (Fiji).

NICD intensity for each cell was measured within the cell, and background mean grey intensity was also measured. Signal intensity was calculated by subtracting the cell area multiplied by the mean grey value of the background from the raw intermediate density (RawIntDen) of the cell. Values of secondary antibody only incubated cells were subtracted from the fully stained intensities to get normalized intensity values. To confirm NICD signal was real, iPS cells were treated with either 150 nM LY411575 or 1 µM MLN4924 or DMSO for 3 h. MLN4924 acted as an amplifier of NICD signal by decreasing its degradation and served as positive control for NICD signal and LY411575 served as a negative control, as it blocked proteolytic cleavage of the intracellular Notch domain.

**Mouse embryo explant culture**

E10.5 CD1 mice embryos were harvested and explants were prepared as previously described [88]. Each embryo's posterior part was divided into two halves by cutting along the neural tube. The explants were cultured in hanging drops of culture medium composed of DMEM/F12 (Gibco), 10% FBS, 1% penicillin–streptomycin, 10 ng/ml Fgf2 (PeproTech).

One half was cultured in medium containing Purvalanol B (1 µM) or RO-3306 (10 µM), whereas the control side was cultured in normal medium (DMSO). Both sides were cultured for 4 h and then analysed for expression of *Lfng* mRNA by *in situ* hybridization.

For the Western blot analysis, PSM explants from 5 embryos were cultured in DMSO control vehicle and the corresponding contralateral 5 PSM explants cultured in the presence of a small molecule inhibitor.

Experiments were conducted in strict adherence to the Animals (Scientific Procedures) Act of 1986 and UK Home Office Codes of Practice for use of animals in scientific procedures.

**In situ hybridization of PSM explants using exonic RNA probes**

For whole-mount embryos and explants, *in situ* hybridization analysis was performed with the use of exonic anti-sense RNA probes as previously described [89].

Samples were imaged using a Leica bright field dissection microscope using Volocity acquisition software. The number of somites present in each explant (identical between explant pairs but variable between embryos) was recorded.

To fix the tissue, samples were incubated for 2 h at RT in 4% PFA/PBS, washed copiously in PBST and transferred into sealed Eppendorf tubes containing 0.01% sodium azide/PBS for long-term storage at 4°C.

**Chicken culture and somite size analysis**

Fertilized chick (*Gallus gallus*) embryos from Henry Stewart & Co farm, Norfolk, UK, were incubated for approximately 40 h at 38°C 5% $CO_2$ to yield embryos between Hamburger Hamilton (HH) stages

9–10 as judged by somite number and morphology. Whole chicken embryos were cultured on filter papers with 1 µM of Purvalanol B or 10 µM of RO-3306 in culture media for 7.5 h. After fixation, embryos were imaged and somite size recorded using ImageJ software.

We define $b_{i,j}$ to be the position of the $j^{th}$ somite boundary in the $i^{th}$ PSM sample where $j = 0$ represents the posterior-most boundary in the embryo at the end of the experiment, $j = 1$ represents the boundary anterior to that etc.

The length of the $j^{th}$ somite is defined to be:

$$S_{i,j} = b_{i,j} - b_{i,j-1}, \, j = 1, 2, 3, \ldots$$

The ratio of successive somite lengths is given by:

$$r_{i,j} = \frac{S_{i,j}}{S_{i,j+1}}, \, j = 1, 2, 3, \ldots$$

The log2 fold change is given by:

$$\log_2(r_{i,j}), \, j = 1, 2, 3, \ldots$$

Samples ($n = 14$) were pooled based upon treatment protocol: DMSO ($i = 1,\ldots,14$), PurB, ($i = 15,\ldots,28$), RO-3006, ($i = 29,\ldots,42$). The mean value at each somite index was computed together with the standard error of the mean. Data at each somite position were tested for deviation from normality using a Kolmogorov–Smirnov test, and the *null* hypothesis was not rejected at the 5% significance level. Paired *t*-tests were used to compare the effect of respective treatments with control (DMSO treatment) at each somite position.

**Statistical analysis**

Unless indicated, statistical analyses were performed using GraphPad software.

One-way ANOVA or Student's *t*-tests were performed in all the data comparing control to treatment conditions, with *P* values calculated as *$P \leq 0.05$, **$P \leq 0.01$ and ***$P \leq 0.001$. All experiments have been performed at least three times.

**Expanded View** for this article is available online.

### Acknowledgements

We would like to thank Ioanna Mastromina and Michaela Omelkova for reading of the manuscript and constructive feedbacks. Special thanks also go to Genta Ito for assistance with Phos-tag gel assay. We thank the groups of D. Alessi and S. Rocha for reagents and Laura D'Ignazio for experimental assistance. The authors would like to thank Rachel Toth, Mel Wightman and Tom Macartney (MRC-PPU Reagents and Services) for cloning and Dr David Campbell, Bob Gourlay and Joby Vhargese (MRC-PPU) for mass spectrometry analysis. We also thank Professor Daan van Aalten and Professor Victoria Cowling and for their inputs. This work was supported by a Wellcome Trust PhD studentship to F.A.C.; an MRC PhD studentship to D.D.; an MRC grant [MC_UU_12016] to P.D.; a Wellcome Trust Strategic award [097945/Z/11/Z] to J.K.D.

### Author contributions

FAC performed all the experiments; designed the figures; generated, analysed and interpreted the data; and drafted the manuscript. PM developed the mathematical model and contributed to the manuscript writing for modelling section.

DD performed experiment shown in Fig 6. MAF generated HEK293T CDK2$^{-/-}$ cells. PD helped supervising the project; contributed to the interpretation of the results; provided financial support; and reviewed the manuscript. JKD devised and supervised the project; responsible for project conception and major contribution to the design of experiments and data interpretation; validated all data; provided financial support; drafted and reviewed the manuscript.

## Conflict of interest

The authors declare that they have no conflict of interest.

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
