## [Review Process File · EMBO Reports]

CDK1 and CDK2 regulate NICD1 turnover and the periodicity of the segmentation clock

Francesca Anna Carrieri, Philip Murray, Dimitrinka Ditsova, Margaret Ashley Ferris, Paul Davies, Jacqueline Kim Dale

Review timeline:

Submission date:	16 May 2018
Editorial Decision:	28 June 2018
Revision received:	9 November 2018
Editorial Decision:	18 December 2018
Revision received:	15 January 2019
Editorial Decision:	28 January 2019
Revision received:	11 March 2019
Accepted:	26 March 2019

Editor: Achim Breiling

Transaction Report:

1st Editorial Decision

28 June 2018

Thank you for the submission of your research manuscript to EMBO reports. We have now received reports from the three referees that were asked to evaluate your study, which can be found at the end of this email.

As you will see, all referees think the manuscript is of interest, but requires further revisions to allow publication in EMBO reports. All three referees have a number of concerns and/or suggestions to improve the manuscript, which we ask you to address in a revised manuscript. As the reports are below, I will not detail them here. In particular, we think that further experimental data in a more *in vivo* context needs to be provided, in order to underline the physiological relevance of the findings, as outlined by referee #1 and #3 (in particular his/her point 5). However, we also think that the final suggestion of reviewer 1 (the use of inducible CDK KO embryos) would be outside the scope of the present manuscript. Further, we ask you to follow the suggestion of referee #2 and to simplify the paper (reducing its length and selecting a subset of panels from each of the main figures), which will greatly improve the readability of the manuscript, and allow many of the comments made by referees #1 and #3 to be addressed.

Given the constructive referee comments, we would like to invite you to revise your manuscript with the understanding that all referee concerns must be addressed in the revised manuscript and in a detailed point-by-point response. Acceptance of your manuscript will depend on a positive outcome of a second round of review. It is EMBO reports policy to allow a single round of revision only and acceptance or rejection of the manuscript will therefore depend on the completeness of your responses included in the next, final version of the manuscript.

Revised manuscripts should be submitted within three months of a request for revision; they will otherwise be treated as new submissions. Please contact us if a 3-months time frame is not sufficient for the revisions so that we can discuss the revisions further.

Supplementary/additional data: The Expanded View format, which will be displayed in the main HTML of the paper in a collapsible format, has replaced the Supplementary information. You can submit up to 5 images as Expanded View. Please follow the nomenclature Figure EV1, Figure EV2 etc. The figure legend for these should be included in the main manuscript document file in a section called Expanded View Figure Legends after the main Figure Legends section. Additional Supplementary material should be supplied as a single pdf labeled Appendix. The Appendix includes a table of content on the first page, all figures and their legends. Please follow the nomenclature Appendix Figure Sx throughout the text and also label the figures according to this nomenclature.

For more details please refer to our guide to authors:
<http://embor.embopress.org/authorguide#manuscriptpreparation>

Important: All materials and methods should be included in the main manuscript file.

See also our guide for figure preparation:
http://www.embopress.org/sites/default/files/EMBOPress_Figure_Guidelines_061115.pdf

Regarding data quantification and statistics, can you please specify, where applicable, the number "n" for how many independent experiments (biological replicates) were performed, the bars and error bars (e.g. SEM, SD) and the test used to calculate p-values in the respective figure legends. Please provide statistical testing where applicable. See:
<http://embor.embopress.org/authorguide#statisticalanalysis>

Please also follow our guidelines for the use of living organisms, and the respective reporting guidelines: <http://embor.embopress.org/authorguide#livingorganisms>

We now strongly encourage the publication of original source data with the aim of making primary data more accessible and transparent to the reader. The source data will be published in a separate source data file online along with the accepted manuscript and will be linked to the relevant figure. If you would like to use this opportunity, please submit the source data (for example scans of entire gels or blots, data points of graphs in an excel sheet, additional images, etc.) of your key experiments together with the revised manuscript. Please include size markers for scans of entire gels, label the scans with figure and panel number, and send one PDF file per figure.

Please also note that we now mandate that all corresponding authors list an ORCID digital identifier that is linked to their EMBO reports account!

I look forward to seeing a revised version of your manuscript when it is ready. Please let me know if you have questions or comments regarding the revision.

REFEREE REPORTS

Referee #1:

In this manuscript, the authors focus in the relevance of Notch1-ICD in somitogenesis and how changes in Notch1-ICD stability can affect this process. In fact they mention a recent study using a pharmacological approach that demonstrate that culturing chick/mouse PSM explants with broad specificity inhibitors leads to elevated levels and a prolonged NICD half-life and phase shifted clock oscillation patterns at a tissue level, leading to larger segments. Furthermore, reducing NICD production in this assay rescues these effects [18].

What is new in the manuscript from Carrieri et al is the identification of CDK1 and CDK2 as kinases that phosphorylate NICD to regulate its stability thus controlling somitogenesis clock and somite formation. However, multiple issues should be addressed to better support authors' conclusions.

First, although in the introduction authors highlight the importance of Notch in somitogenesis, most

of the work presented here is done in HEK293T cells that are primarily used in overexpression experiments but do not represent the best model for understanding physiological Notch regulation. In addition, it is not known whether in these cells differences in confluence, density at seeding, or differential proliferation after inhibitor treatment can directly affect NICD levels. It is also unknown which are the ligands that induce Notch1 activation in these cells and whether Notch ligands can be affected by the inhibitors used or by the culture conditions.

Also, there are specific inconsistencies in the manuscript that should be addressed or discussed.

Figures 1A and 1E are redundant but results seem rather different.

In Figure 3 it is stated that Phosphorylation of serine 2513, but not serine 2516, is required for the NICD-FBW7. However, this assumption is based in the observation that mutation of S2513 into alanine prevent Fbw7 co-precipitation what is suggestive but not conclusive.

In Figure 4, CDK1 depletion imposes a higher effect in NICD stabilization compared with CDK2 KO, even when CDK2 levels increases in this conditions. Moreover, these results are in agreement with the higher phosphorylation of the peptide containing S2513 by CDK1 compared with CDK2. However, subsequent analysis mainly focuses on CDK2. Is there any particular reason for taking this apparently arbitrary decision?

In 6A and 6B changes in NICD levels upon CDK2 depletion are completely different (high increase in the CDK2^{-/-} in 6A and only minimal in 6B where double KO is more similar to single KO in A).

In 7C it is shown that NICD levels fluctuate during cell cycle in a phase that may suggest a CDK-dependent regulation. However, cell cycle analysis shown in Figures 7D and 7E after CDK2 depletion are marginal. Authors should show the analysis of the double single CDK1 and double CDK1/2 KO and also the western blot analysis shown in 7C but using the CDK1 and CDK2 KO cells.

Then in Figure 8 authors move to the iPSC model to show IF of NICD in cells at different phases of the cell cycle and what authors conclude is that NICD levels peak at G1 and G2 phases of the cell cycle. However, if every dot represents a single cell counted, differences between G1/2 and S at mainly due to few out-layers with very high Notch1 staining.

In Figure 9 they focus in the use of Purvalanol B a selective inhibitor of CDK2 (they could also check Roscovitine and GSK650... that are very active against this same kinase) and they co-treat with MLN to increase Notch levels. However, no changes in the electrophoretic mobility of NICD are shown after CDK2 inhibition. Then authors say that only the upper band of NICD coprecipitates with FBW7 suggesting that phosphorylation is important for Notch and Fbw7 interaction, but in that case CDK2 independent. In 9C and 9E, levels of NICD in embryos treated with PurB or Ro-3306 need to be quantified, as differences are maginal. Again, no differences in electrophoretic mobility are seen. Analysis of the embryos treated with CDK2 or CDK1 inhibitors, which to my view represent the most relevant part of the work is only anecdotic.

To my view, if authors want to construct a relevant story linking CDK activity during cell cycle with NICD stability and function associated with somitogenesis much work need to be done specifically in the embryo-related part of the study. For example, authors could use inducible CDK KO embryos, test different inhibitors and combinations of them, check different markers other than Lnfg and finally perform a more accurate and quantitative analysis of the phenotypes observed, and demonstrate whether they are Notch-dependent or not. Moreover, Notch1 mutants in the serines that are targeted by CDK1/2 need to be further tested for their protein-stability dependent of CDK activity, Notch regulation by cell cycle in the presence or absence of CDK1/2 and their possible contribution to cell cycle regulation.

Referee #2:

This is a detailed, well executed study by Carrieri et al which demonstrates that CDK1 and 2

regulate NICD1 turnover and periodicity of the segmentation clock. The authors provide convincing experimental evidence using both in vitro and in vivo approaches. They identify a crucial residue involved in NICD recognition by SCF E3 ligase, show that both kinases phosphorylate NICD in the region where this residue is located, and intriguingly show that NICD levels vary in a cell cycle dependent manner. These novel data will be of great interest, not only to people working directly in the Notch field but to other biologists working in systems where the Notch signal plays an important regulatory role.

It should be noted that I have not assessed the mathematical model as it is outside my expertise.

The authors take a thorough approach and provide a range of supporting data for their observations. In some cases it would help the reader if the figures in the main body of the manuscript could be simplified and some panels removed to supplementary (eg Fig 2 Retain A-D, Move parts E,F,G to Suppl. This will not lessen their impact but it will make it easier for the reader to appreciate the thorough approach).

Minor points

Line 49 -list the 4 genes involved in SCD

Line 68- Upon extracellular ligand binding

Line 97- substitute "anti-correlating" with "correlating inversely" if this is what is meant

Line 117- It would be helpful to have some idea of what levels of Notch and NICD are present in these cell lines to explain why they were selected, is it a range of levels low to high? And is much difference in sensitivity to the inhibitors seen?

Line 173- I am not familiar with phos-tag technology, why are the phospho-bands in 1E so numerous compared to 1A, is it sensitivity or a different resolution, a brief comment would help the reader.

Line 186 After treatment with MLN4924, insert "an inhibitor of"

Line 209 Justify briefly use of the HEK cell for the mass spec analysis

Line 224 suggest removing lines 224-234 and just include references as this section is too long.

Line 293 What conditions have been used to test inhibitory potential of eg Roscovitine (see Figure 5)? Are these values derived for cell culture models or a different type of assay?

Discussion could be shortened considerable (~a third), it reads long.

I suggest a final summary figure should be included which has a cartoon or simple outline of the links shown between NICD, phosphorylation and the cell cycle in this paper. This would nicely round off this multidisciplinary study.

----- Referee #3:

The manuscript from Carrieri et al, follows up on previously published work from this lab that demonstrated that several small molecule inhibitors affected the half life and level of NICD with the effect of lengthening the period of the segmentation clock. In this manuscript, the authors further examine the regulatory mechanism(s) that contribute to this effect, suggesting that some NICD phosphorylation states influence association with the SCF E3 ligase component FBXW7, protecting NICD from degradation and prolonging its half life. The authors go on to examine a link between CDK phosphorylation and NICD turnover and identify cell cycle linked patterns in NICD in tissue culture cells. Experiments using cultured mouse PSMs suggest that phosphorylation of NICD by CDKs may influence the period of the segmentation clock and mathematical modeling works to integrate these findings into what we know about Notch signaling in this context,

Overall, these findings may be important on many levels. Post-transcriptional regulation of the Notch pathway is critically important for many developmental decisions and in disease states including cancers. Understanding how NICD turnover is regulated will be of wide interest. Specific post-transcriptional mechanisms that regulate the segmentation clock are poorly understood, and are important for the development and evolution of the axial skeleton, and have implications for other situations where oscillatory Notch signaling may be important.

Several issues should be addressed to robustly test the hypothesis and strengthen the rigor of the conclusions prior to publication:

MAJOR POINTS

1. Statistics: At many points in the manuscript where multiple pairwise comparisons are made, statistical analysis would more appropriately be done as a one-way ANOVA analysis followed by a post hoc tests for individual significance to control for multiple testing. This should be done at a minimum for 1B, 1D, 5B, 8B, And Supp1B
2. In Figure 1B and 1D, it is not clear to me why treatment with phosphatase also changes (reduces) the total amount of protein, rather than just its mobility (ie, why are the proteins levels quantified in the MLN4924+lpp treatment conditions the same as levels seen with DMSO? I would anticipate that phosphatase treatment would simply collapse the two bands, leading to a darker lower band, that was still more intense than in the DMSO treated cells once normalization was completed). This should be explained at some point.
3. Regarding experiments demonstrating an effect on NICD half life/turnover: The figure legend for Supplemental Figure 2 does not seem to match the description in the text. The legend says that all lanes were treated with the labeled inhibitors for 3 hours and for the last hour of culture with LY411575. This would suggest that the samples in lane 2 was also treated for the full time in culture with LY411575 (supported by the statement in the legend that "HEK293 cells were treated for 3 hours with 150 nM of LY411575". If this is correct, the low levels of NICD in lane 2 are due to chronic, 3 hour inhibition of Notch by LY411575, and changes in stability would be seen by comparing the levels of NICD in lane 1 (3 hours DMSO, 1 hour LY411575) to NICD in lanes 3-7 (3 hours inhibitor + 1 hour LY411575). If the figure labeling and legend are correct, the increases in NICD in lanes 3-7 are not robustly convincing compared to the levels of NICD in lane 1. The description on page 7 seems to indicate that lane one was not treated with LY411575 at all, and that lane 2 was treated with DMSO for 3 hours with LY411575 treatment only in the final hour. If the test is correct, the figure and legend need to be altered. If the figure legend and labeling are correct, additional work is needed to convincingly demonstrate that changes in NICD stability are the cause of increases NICD levels, including replication and quantification of NICD to demonstrate significance.
4. The central model for how the inhibitors are altering NICD turnover could be more clearly stated and demonstrated. My reading was that the authors suggest that 1) CDK-mediated phosphorylation of NICD at S2413 promotes (or is required for) NICD interactions with FBXW7 that promote degradation 2) The inhibitors examined here and in the Wiedemann paper act to prevent that phosphorylation event (either by altering the phosphorylation profile of NICD or by inhibiting relevant CDKs. I still feel that some internal steps in the logical progression are missing somehow.
 - a) In figure 1E, treatment with MLN4924 leads to accumulation of several different phosphorylated versions of NICD. Treatment with different inhibitors leads to accumulation of different subsets of phosphorylated NICD, which are presumably more stable? or protected from degradation somehow?
 - b) In Fig 1E the DMSO lysates appear to have faint bands at the lowest band size and at least at one of the upper band sizes. An overexposed image of this region of the gel would help demonstrate whether the inhibitors are changing the ratios of lower to upper bands, or simply stabilizing all of the bands equally.
 - c) Phosphatase treatment of inhibitor treated lysates would demonstrate that the isoforms seen after inhibitor treatment actually collapse would be interesting, since one of the points is that these are distinct phosphorylation events from those that are accumulating when SCF ligase is inhibited.
 - d) Concomitant treatment with various inhibitors + MLN4924 followed by Phos-tag would demonstrate whether the inhibitors are actually preventing the production of the uppermost phosphorylated band of NICD that accumulates after MLN inhibition but not after treatment with other inhibitors, giving hints about mechanisms
 - e) Phos-tag analysis of lysates after treatment with specific CDK1 and CDK2 inhibitors would be informative.
5. The data in figure 9 that specific inhibition of CDK1 or CDK2 lengthens the clock period are unconvincing as presented. Additional support minimally in the form of additional images demonstrating differences between treated and untreated half PSMs allowing some sort of understanding of how significant or variable the delays are is needed. Additional possibilities that would strengthen these conclusions might include: wholemount IHC for NICD in treated and untreated half tails to give an idea of whether NICD patterns are being altered in addition to NICD levels and/or experiments measuring somite sizes in treated and untreated explants similar to those done in the Wiedemann paper.

MINOR POINTS

1. I have some concerns with clarity of writing throughout - for instance pg 3 line 44 refers to "these clock genes" when clock genes have not clearly been defined. Careful reading by the authors for clarity will be important.
2. Line 49 - SCDO is caused by mutations in at least 6 genes DLL3, LFNG, HES7, MESP2, TBX6, AND RIPPLY2. While DLL3 and LFNG are clearly members of the Notch pathway, and HES7 (and perhaps MESP2) could be described as pathway targets, the statements that "four genes lead to familial forms of SCD [8]. Significantly, these are components of the Notch pathway" is not particularly clear.
3. Line 72 what is meant by the idea that "most canonical Notch" activity relies on turnover? Does this mean is regulated via turnover?
4. While the authors are correct on page 3, line 61 that little work examining the effects of component stability on clock period have been performed, several groups have examined the importance of component half lives for clock function. The authors may wish to acknowledge some of this previous work.
5. Page 8 lines 178 - 181 the statements "NICD-FBXW7 interact at the endogenous levels in HEK293 cells" and "the NICD-FBXW7 interaction has only been shown in overexpressed systems" seem contradictory (although perhaps the first statement is meant to be a bold heading describing the conclusion of this section?)
6. Figure 2B - Which bands are being quantified? are the upper and lower bands being quantified together? Also, the difference in NICD levels between DMSO and MLN treated cells seems much more dramatic in 2A than the quantification in 2B would suggest - is 2A a representative blot?
7. In figure 7c, there seems to be a loss of NICD at 12 hour timepoint as well - what is the suggested explanation for this?
8. page 20 line 496 - I don't understand the implication here "In each case, however, the highest molecular bands were no longer visible." The highest molecular weight bands are only seen when proteasome inhibition is done. "no longer seen" would imply that if you simultaneously treat with (for example) Roscovitine and MLN, you would not see the highest bands seen with MLN treatment alone. In fact, that condition might help tease apart whether the phosphorylation events you get after inhibitor treatments are, in fact, protecting the NICD from being phosphorylated in ways that contribute to the highest molecular weight bands seen.
9. page 21, line 510 the statement "when mammalian Sel-10 (homologue of FBXW7 in C.elegans) is mutated, NICD is much more stable" Isn't Sel-10 a C elegans protein?
10. Page 24 line 587, the authors state "It is noteworthy that inhibition of CDK2 activity with a highly selective inhibitor reduces the NICD-FBXW7 interaction in HEK293 cells but does not block it completely, as we saw with Roscovitine and DRB." I initially read this as suggesting that ROS and DRB DO block completely However, the quantification in 9B shows a 80% reduction in what I assume are fold changes in NICD levels after MLN+PurB treatment, which seems quite comparable to the quantified reduction in Figure 2 (which look to me like about 80% in MLN+ROS and 85% in MLN+DRB). Or does the author mean "as we saw" to suggest a similar effect in the three conditions?
11. Label the axis in 9B, 10B
12. Include methods for quantification of western blots, including how it was determined that exposures were within the linear range.

1st Revision - authors' response

9 November 2018

Referee #1:

In this manuscript, the authors focus in the relevance of Notch1-ICD in somitogenesis and how changes in Notch1-ICD stability can affect this process. In fact they mention a recent study using a pharmacological approach that demonstrate that culturing chick/mouse PSM explants with broad specificity inhibitors leads to elevated levels and a prolonged NICD half-life and phase shifted clock oscillation patterns at a tissue level, leading to larger segments. Furthermore, reducing NICD production in this assay rescues these effects [18].

What is new in the manuscript from Carrieri et al is the identification of CDK1 and CDK2 as kinases that phosphorylate NICD to regulate its stability thus controlling somitogenesis clock and

somite formation. However, multiple issues should be addressed to better support authors' conclusions.

First, although in the introduction authors highlight the importance of Notch in somitogenesis, most of the work presented here is done in HEK293T cells that are primarily used in overexpression experiments but do not represent the best model for understanding physiological Notch regulation.

We thank the referee for these comments. We would like to point out that we did not use HEK293 cells for overexpression only. In fact, our assays focussed predominantly on analysing endogenous NICD. We also tested the effect of these reagents on levels of endogenous NICD in iPS cells, IMR90 fibroblasts and in addition, in this re-submission, we have included an analysis in mES cells, all of which responded in the same way suggesting the pathway is universal in its degradation pathway.

In addition, it is not known whether in these cells differences in confluence, density at seeding, or differential proliferation after inhibitor treatment can directly affect NICD levels.

All cell plates for each experimental unit were seeded from the same base stock and were of equivalent confluency before treatment. Protein levels were normalised by Bradford assay and relative differences in the levels of control proteins were used to normalise differences in NICD total and phospho-levels.

It is also unknown which are the ligands that induce Notch1 activation in these cells and whether Notch ligands can be affected by the inhibitors used or by the culture conditions.

In all of the assays we conduct, we are looking at what happens intracellularly, post-ligand interaction. Moreover, the assay in Appendix S2B shows that the effects we are looking at are not due to increased NICD production (ligand dependent) but due to NICD stability (ligand independent).

Figures 1A and 1E are redundant but results seem rather different.

We thank the referee for these comments. Figures 1A and 1E are data from two different analyses: figure 1A is a Western blot which will highlight isoforms of the NICD protein that do or do not carry post-translational modifications such as phosphorylated residues. Figure 1E is a Phos-tag gel assay which will specifically show isoforms of the NICD protein that have different molecular weights due to the different number of phosphorylated residues (please see Ito, G., Tomita, T. Rab10 Phosphorylation Detection by LRRK2 Activity Using SDS-PAGE with a Phosphate-binding Tag. *J. Vis. Exp.* (130), e56688, doi:10.3791/56688 (2017).

In addition, we added a sentence in the main text to clarify what the Phos-tag assay represents.

In Figure 3 it is stated that Phosphorylation of serine 2513, but not serine 2516, is required for the NICD-FBXW7. However, this assumption is based in the observation that mutation of S2513 into alanine prevent Fbw7 co-precipitation what is suggestive but not conclusive.

We agree with the reviewer and have made the change accordingly.

In Figure 4, CDK1 depletion imposes a higher effect in NICD stabilization compared with CDK2 KO, even when CDK2 levels increases in this conditions. Moreover, these results are in agreement with the higher phosphorylation of the peptide containing S2513 by CDK1 compared with CDK2. However, subsequent analysis mainly focuses on CDK2. Is there any particular reason for taking this apparently arbitrary decision?

We would respectfully argue we focussed on both CDKs in this manuscript. We would like to highlight that we used both siRNA and pharmacological inhibition of both CDK1 and CDK2 in the cell assays and pharmacological inhibition of both CDK1 and CDK2 in the embryo assays. We did additionally use a CDK2^{-/-} line for the cell assays. We would have used a KO of both kinases but this is not possible while also maintaining healthy, dividing cells. Nevertheless, we then used this CDK2^{-/-} tool and additionally removed CDK1 activity transiently with siRNA.

In 6A and 6B changes in NICD levels upon CDK2 depletion are completely different (high increase in the CDK2^{-/-} in 6A and only minimal in 6B where double KO is more similar to single KO in A).

Data showed in Fig 6A and 6B come from two independent experiments, therefore one can not compare them. Moreover, the quantifications are now provided for the blots in Figure 6 (Figures 6B and D).

In 7C it is shown that NICD levels fluctuate during cell cycle in a phase that may suggest a CDK-dependent regulation. However, cell cycle analysis shown in Figures 7D and 7E after CDK2 depletion are marginal.

We would respectfully argue there is a small but nevertheless statistically significant increase in the number of cells in G1 and a decrease in the number of cells in S phase after CDK2 depletion, as expected given the function of this kinase in the cell cycle (Figure 7C).

Authors should show the analysis of the single CDK1 and double CDK1/2 KO and also the western blot analysis shown in 7C (now 7B) but using the CDK1 and CDK2 KO cells.

We thank the reviewer for this observation. We performed an additional experiment (FACS analysis after CDK1 siRNA depletion) and we observe a marginal but nevertheless significant accumulation of cells in G2 phase compared to the control, as expected for cells deprived of CDK1 activity and therefore unable to pass the G2/M checkpoint (Figure 7D). As requested we also conducted the western blot analysis shown in 7B but using the CDK2 KO cells (Appendix S5D).

We were not able to performed the western blot analysis on the double CDK1/2 KO cells for experimental limitations, combining double thymidine block and release assay with siRNA-depletion protocols.

Then in Figure 8 authors move to the iPSC model to show IF of NICD in cells at different phases of the cell cycle and what authors conclude is that NICD levels peak at G1 and G2 phases of the cell cycle. However, if every dot represents a single cell counted, differences between G1/2 and S at mainly due to few out-layers with very high Notch1 staining.

We thank the reviewer for this observation. We repeated the experiment to increase the number of cells analysed. The data in Figure 8 now very clearly shows this difference in levels of NICD is robustly significant.

In Figure 9 they focus in the use of Purvalanol B a selective inhibitor of CDK2 (they could also check Roscovitine and GSK650... that are very active against this same kinase) and they co-treat with MLN to increase Notch levels. However, no changes in the electrophoretic mobility of NICD are shown after CDK2 inhibition. Then authors say that only the upper band of NICD coprecipitates with FBW7 suggesting that phosphorylation is important for Notch and Fbw7 interaction, but in that case CDK2 independent.

We have already shown a reduction in the Notch and Fbxw7 interaction in the presence of Roscovitine (Figure 2A-B), DRB (Appendix S3A-B) and Purvalanol (Figure 9A-B). In addition, we performed immunoprecipitation experiment testing the whether GSK650 affects the NICD-Fbxw7 interaction (Appendix S3C-D).

We show clearly in Figure 3 there are numerous residues in NICD that are phosphorylated and that there are numerous phosphorylated isoforms of NICD (Figure 1E and Appendix S2B).

We show, using Phos-tag gels, Roscovitine and Purvalanol B, reduce some of these phosphorylation events but not all of them (Figure 1E and Appendix S2B). Thus, there will not necessarily be an obvious shift in electrophoretic mobility by standard Western blot analysis i.e. there is still an upper and lower band in the input lanes of blots shown in Figures 2 and 9 after Rosc/PurB treatments, as there are still phosphorylated isoforms of NICD that are CDK2-independent.

However, the intensity of the bands in the IP lanes is significantly reduced – thus the phosphorylation conducted by the kinases that are targeted by Roscovitine and Purvalanol B are important for the Notch and Fbw7 interaction.

In 9C and 9E, levels of NICD in embryos treated with PurB or Ro-3306 need to be quantified, as differences are maginal. Again, no differences in electrophoretic mobility are seen. Analysis of the

embryos treated with CDK2 or CDK1 inhibitors, which to my view represent the most relevant part of the work is only anecdotic.

We thank the reviewer for this comment and have now quantified the effect. See Appendix S6A-D. We do not expect to necessarily observe a change in electrophoresis mobility, considering that PurB or RO3306 may only inhibit one phosphosite, and as we have shown in Figure 1E there are multiple phosphospecies of NICD. We also have added additional images showing the effect on somite formation in mouse embryos (Appendix S6E), as well as new experiments analysing the effect of these inhibitors on chicken embryos (See Figures 9E-F).

To my view, if authors want to construct a relevant story linking CDK activity during cell cycle with NICD stability and function associated with somitogenesis much work need to be done specifically in the embryo-related part of the study. For example, authors could use inducible CDK KO embryos, test different inhibitors and combinations of them, check different markers other than Lnfg and finally perform a more accurate and quantitative analysis of the phenotypes observed, and demonstrate whether they are Notch-dependent or not. Moreover, Notch1 mutants in the serines that are targeted by CDK1/2 need to be further tested for their protein-stability dependent of CDK activity, Notch regulation by cell cycle in the presence or absence of CDK1/2 and their possible contribution to cell cycle regulation.

Following editorial advise, we did not need to address this comment as it is out with the remit of this study.

Referee #2:

This is a detailed, well executed study by Carrieri et al which demonstrates that CDK1 and 2 regulate NICD1 turnover and periodicity of the segmentation clock. The authors provide convincing experimental evidence using both in vitro and in vivo approaches. They identify a crucial residue involved in NICD recognition by SCF E3 ligase, show that both kinases phosphorylate NICD in the region where this residue is located, and intriguingly show that NICD levels vary in a cell cycle dependent manner. These novel data will be of great interest, not only to people working directly in the Notch field but to other biologists working in systems where the Notch signal plays an important regulatory role.

It should be noted that I have not assessed the mathematical model as it is outside my expertise.

The authors take a thorough approach and provide a range of supporting data for their observations. In some cases it would help the reader if the figures in the main body of the manuscript could be simplified and some panels removed to supplementary (eg Fig 2 Retain A-D, Move parts E,F,G to Suppl . This will not lessen their impact but it will make it easier for the reader to appreciate the thorough approach).

We thank the reviewer for this suggestion and have performed this action (see Figure 2 and Appendix S3).

Minor points

Line 49 -list the 4 genes involved in SCD completed

Line 68- Upon extracellular ligand binding completed

Line 97- substitute "anti-correlating" with "correlating inversely" if this is what is meant completed

Line 117- It would be helpful to have some idea of what levels of Notch and NICD are present in these cell lines to explain why they were selected, is it a range of levels low to high? And is much difference in sensitivity to the inhibitors seen?

We selected HEK293 cells as they are widely used for biochemical assays, levels of NICD were comparatively good in this cell line and they are easy to work with; we chose iPS and mES cells to have a more physiological context, we chose IMR90 fibroblast cells as they are embryonic and genomically stable, compared to HEK293 cells. All cell lines selected showed similar sensitivity to these inhibitors suggesting it is a universal mechanism of regulating NICD turnover.

Line 173- I am not familiar with phos-tag technology, why are the phospho-bands in 1E so numerous compared to 1A, is it sensitivity or a different resolution, a brief comment would help the

reader. We thank the reviewer for this suggestion and we have added a brief description and a reference in the main text : Phos-tag is a modified western blot assay which separates out and therefore allows simultaneous visualisation of different phosphorylated isoforms of a given protein of interest, as a result of their different migration speeds.

Line 186 After treatment with MLN4924, insert "an inhibitor of" completed

Line 209 Justify briefly use of the HEK cell for the mass spec analysis

We selected HEK293 as a cell model to perform Mass Spectrometry experiments for all of the reasons already reported above and because it is widely used for this technique and finally because all our previous experiments have been optimised in this cell line.

Line 224 suggest removing lines 224-234 and just include references as this section is too long. completed

Line 293 What conditions have been used to test inhibitory potential of eg Roscovitine (see Figure 5)? Are these values derived for cell culture models or a different type of assay?

The values come from the world leading MRC Kinase Profiling Inhibitor Database (<http://www.kinase-screen.mrc.ac.uk/screening-compounds/345922>).

Discussion could be shortened considerable (~a third), it reads long.

We thank the reviewer for this suggestion and have performed this action.

I suggest a final summary figure should be included which has a cartoon or simple outline of the links shown between NICD, phosphorylation and the cell cycle in this paper. This would nicely round off this multidisciplinary study.

We thank the reviewer for this suggestion and have performed this action (see Appendix S7).

Referee #3:

The manuscript from Carrieri et al, follows up on previously published work from this lab that demonstrated that several small molecule inhibitors affected the half life and level of NICD with the effect of lengthening the period of the segmentation clock. In this manuscript, the authors further examine the regulatory mechanism(s) that contribute to this effect, suggesting that some NICD phosphorylation states influence association with the SCF E3 ligase component FBXW7, protecting NICD from degradation and prolonging its half life. The authors go on to examine a link between CDK phosphorylation and NICD turnover and identify cell cycle linked patterns in NICD in tissue culture cells. Experiments using cultured mouse PSMs suggest that phosphorylation of NICD by CDKs may influence the period of the segmentation clock and mathematical modeling works to integrate these findings into what we know about Notch signaling in this context,

Overall, these findings may be important on many levels. Post-transcriptional regulation of the Notch pathway is critically important for many developmental decisions and in disease states including cancers. Understanding how NICD turnover is regulated will be of wide interest. Specific post-transcriptional mechanisms that regulate the segmentation clock are poorly understood, and are important for the development and evolution of the axial skeleton, and have implications for other situations where oscillatory Notch signaling may be important.

Several issues should be addressed to robustly test the hypothesis and strengthen the rigor of the conclusions prior to publication:

MAJOR POINTS

1. Statistics: At many points in the manuscript where multiple pairwise comparisons are made, statistical analysis would more appropriately be done as a one-way ANOVA analysis followed by a post hoc tests for individual significance to control for multiple testing. This should be done at a minimum for 1B, 1D, 5B, 8B, And Supp1B

We thank the reviewer for this suggestion and have performed this action (see Figures 1B, 1D, 5C, 8B, and Appendix S1B).

2. In Figure 1B and 1D, it is not clear to me why treatment with phosphatase also changes (reduces) the total amount of protein, rather than just its mobility (ie, why are the proteins levels quantified in the MLN4924+lpp treatment conditions the same as levels seen with DMSO? I would anticipate that phosphatase treatment would simply collapse the two bands, leading to a darker lower band, that was still more intense than in the DMSO treated cells once normalization was completed). This should be explained at some point.

We thank the reviewer for this observation. There reason the levels seem to drop with +lpp is because of saturation on the blot. The MLN conditions are both maxed out in signal intensity so any shift of the phospho-species back into the non phosband is lost. It is also important to remember that the western blots are a representation of at least three independent experiments, whereas the quantifications include all of the biological reps.

3. Regarding experiments demonstrating an effect on NICD half life/turnover: The figure legend for Supplemental Figure 2 does not seem to match the description in the text. The legend says that all lanes were treated with the labeled inhibitors for 3 hours and for the last hour of culture with LY411575. This would suggest that the samples in lane 2 was also treated for the full time in culture with LY411575 (supported by the statement in the legend that "HEK293 cells were treated for 3 hours with 150 nM of LY411575". If this is correct, the low levels of NICD in lane 2 are due to chronic, 3 hour inhibition of Notch by LY411575, and changes in stability would be seen by comparing the levels of NICD in lane 1 (3 hours DMSO, 1 hour LY411575) to NICD in lanes 3-7 (3 hours inhibitor + 1 hour LY411575). If the figure labeling and legend are correct, the increases in NICD in lanes 3-7 are not robustly convincing compared to the levels of NICD in lane 1. The description on page 7 seems to indicate that lane one was not treated with LY411575 at all, and that lane 2 was treated with DMSO for 3 hours with LY411575 treatment only in the final hour. If the test is correct, the figure and legend need to be altered. If the figure legend and labeling are correct, additional work is needed to convincingly demonstrate that changes in NICD stability are the cause of increases NICD levels, including replication and quantification of NICD to demonstrate significance.

We thank the reviewer for this important observation. The legend, text and figure were not coordinated and we have re-performed this assay such that all lanes were treated with the labelled inhibitors for 3 hours and for the last hour of culture with LY411575. The data now convincingly demonstrate that changes in NICD stability are the cause of increases NICD levels.

4. The central model for how the inhibitors are altering NICD turnover could be more clearly stated and demonstrated. My reading was that the authors suggest that 1) CDK-mediated phosphorylation of NICD at S2413 promotes (or is required for) NICD interactions with FBXW7 that promote degradation 2) The inhibitors examined here and in the Wiedemann paper act to prevent that phosphorylation event (either by altering the phosphorylation profile of NICD or by inhibiting relevant CDKs. I still feel that some internal steps in the logical progression are missing somehow. a) In figure 1E, treatment with MLN4924 leads to accumulation of several different phosphorylated versions of NICD. Treatment with different inhibitors leads to accumulation of different subsets of phosphorylated NICD, which are presumably more stable? or protected from degradation somehow? b) In Fig 1E the DMSO lysates appear to have faint bands at the lowest band size and at least at one of the upper band sizes. An overexposed image of this region of the gel would help demonstrate whether the inhibitors are changing the ratios of lower to upper bands, or simply stabilizing all of the bands equally. c) Phosphatase treatment of inhibitor treated lysates would demonstrate that the isoforms seen after inhibitor treatment actually collapse would be interesting, since one of the points is that these are distinct phosphorylation events from those that are accumulating when SCF ligase is inhibited.

We thank the reviewer for this comment. Yes as the reviewer surmises we hypothesise that treatment with different inhibitors leads to accumulation of different subsets of phosphorylated NICD, which are more stable as they lack the phosphorylation which would target them for degradation. We have performed the additional experiment suggested here and observe exactly what the reviewer predicted - the isoforms seen after inhibitor treatment actually collapse following phosphatase treatment of inhibitor-treated lysates (Appendix S2B).

d) Concomitant treatment with various inhibitors + MLN4924 followed by Phos-tag would demonstrate whether the inhibitors are actually preventing the production of the uppermost phosphorylated band of NICD that accumulates after MLN inhibition but not after treatment with other inhibitors, giving hints about mechanisms

We thank the reviewer for this comment and we have performed the additional experiment suggested here and observe exactly what the reviewer predicted - concomitant treatment with various inhibitors, we did not see the highest bands seen with MLN treatment alone (Appendix S2B).

e) Phos-tag analysis of lysates after treatment with specific CDK1 and CDK2 inhibitors would be informative.

We thank the reviewer for this comment and we have performed the additional experiment suggested here (Appendix S2B).

5. The data in figure 9 that specific inhibition of CDK1 or CDK2 lengthens the clock period are unconvincing as presented. Additional support minimally in the form of additional images demonstrating differences between treated and untreated half PSMs allowing some sort of understanding of how significant or variable the delays are is needed.

We thank the reviewer for this suggestion and have performed this action (see Appendix S6Ea-d)

Additional possibilities that would strengthen these conclusions might include: wholemount IHC for NICD in treated and untreated half tails to give an idea of whether NICD patterns are being altered in addition to NICD levels and/or experiments measuring somite sizes in treated and untreated explants similar to those done in the Wiedermann paper.

We thank the reviewer for this suggestion and have performed the experiment to address the effect of CDK inhibition on somite size in chicken embryos (see Figure 9E-F).

MINOR POINTS

1. I have some concerns with clarity of writing throughout - for instance pg 3 line 44 refers to "these clock genes" when clock genes have not clearly been defined. Careful reading by the authors for clarity will be important.

We thank the reviewer for this comment and have rectified in the text.

2. Line 49 - SCDO is caused by mutations in at least 6 genes DLL3, LFNG, HES7, MESP2, TBX6, AND RIPPLY2. While DLL3 and LFNG are clearly members of the Notch pathway, and HES7 (and perhaps MESP2) could be described as pathway targets, the statements that "four genes lead to familial forms of SCD [8]. Significantly, these are components of the Notch pathway" is not particularly clear.

We thank the reviewer for this comment and have rectified in the text.

3. Line 72 what is meant by the idea that "most canonical Notch" activity relies on turnover? Does this mean is regulated via turnover?

We thank the reviewer for this comment and have rectified in the text.

4. While the authors are correct on page 3, line 61 that little work examining the effects of component stability on clock period have been performed, several groups have examined the importance of component half lives for clock function. The authors may wish to acknowledge some of this previous work.

We thank the reviewer for this comment and have rectified in the text.

5. Page 8 lines 178 - 181 the statements "NICD-FBXW7 interact at the endogenous levels in HEK293 cells" and "the NICD-FBXW7 interaction has only been shown in overexpressed systems" seem contradictory (although perhaps the first statement is meant to be a bold heading describing the conclusion of this section?)

We thank the reviewer for this comment – the first statement was supposed to be a title in bold for the sub-section and so we have rectified in the text.

6. *Figure 2B - Which bands are being quantified? are the upper and lower bands being quantified together?* Yes, both bands are being quantified.

Also, the difference in NICD levels between DMSO and MLN treated cells seems much more dramatic in 2A than the quantification in 2B would suggest - is 2A a representative blot? Yes, 2A is representative of 3 separate experiments as detailed in the caption 2B.

7. *In figure 7c, there seems to be a loss of NICD at 12 hour timepoint as well - what is the suggested explanation for this?*

The transition from 10h to 12h reflects transition from early to late G1 as shown in Appendix S5A, which would reflect initiation of a new cell cycle with increased CDK2 activity with subsequent expected drop in levels of NICD.

8. *page 20 line 496 - I don't understand the implication here "In each case, however, the highest molecular bands were no longer visible." The highest molecular weight bands are only seen when proteasome inhibition is done. "no longer seen" would imply that if you simultaneously treat with (for example) Roscovitine and MLN, you would not see the highest bands seen with MLN treatment alone. In fact, that condition might help tease apart whether the phosphorylation events you get after inhibitor treatments are, in fact, protecting the NICD from being phosphorylated in ways that contribute to the highest molecular weight bands seen.*

We thank the reviewer for this comment and we have performed the additional experiment suggested here and observe exactly what the reviewer predicted - adding Roscovitine and MLN, we did not see the highest bands seen with MLN treatment alone (Appendix S2B).

9. *page 21, line 510 the statement "when mammalian Sel-10 (homologue of FBXW7 in C.elegans) is mutated, NICD is much more stable" Isn't Sel-10 a C elegans protein?*

We thank the reviewer for this comment and have rectified in the text.

10. *Page 24 line 587, the authors state "It is noteworthy that inhibition of CDK2 activity with a highly selective inhibitor reduces the NICD-FBXW7 interaction in HEK293 cells but does not block it completely, as we saw with Roscovitine and DRB." I initially read this as suggesting that ROS and DRB DO block completely However, the quantification in 9B shows a 80% reduction in what I assume are fold changes in NICD levels after MLN+PurB treatment, which seems quite comparable to the quantified reduction in Figure 2 (which look to me like about 80% in MLN+ROS and 85% in MLN+DRB). Or does the author mean "as we saw" to suggest a similar effect in the three conditions?*

We thank the reviewer for this comment and we have rectified.

11. *Label the axis in 9B, 10B*

We thank the reviewer for this comment and we have rectified.

12. *Include methods for quantification of western blots, including how it was determined that exposures were within the linear range.*

We thank the reviewer for this comment, we have included this information in the methods section and we have added evidence of the fact exposures are in the linear range (see Appendix S1D).

2nd Editorial Decision

18 December 2018

Thank you for the submission of your revised manuscript to our editorial offices. We have now received the reports from the three referees that were asked to re-evaluate your study, which you will find below. As you will see, referee #2 now supports the publication of the manuscript in EMBO reports. Referee #3 has some remaining concerns s/he thinks could be fixed during a final revision, most likely without further experimentation. However, referee #1 has still major concerns. After cross-commenting, referee #1 indicates that minimally these points need to be addressed:

1) There is no evidence that CDK inhibitors do not affect full-length Notch1, subsequently leading

to increased levels of the processed form. It is necessary to show levels of full-length Notch to clarify this issue.

2) Figure S2B does not demonstrated that the effects of CK inhibition on Notch are ligand independent as the Notch-processing inhibitor (LY411575) is added only 1 hour before collecting the lysates but 2 hours after CK inhibitors treatment. One could argue that levels of ICN1 previous to LY incubation are already increased and the differences maintained 1 hour later. The only way to demonstrate that the effect of CDK inhibitors is on stability instead of processing is looking at protein decay in a time course experiment.

3) In the WB analysis of synchronized HEK and CDK2 KO cells, it is stated that there is a dramatic decrease in fluctuations in the KO cells when the only difference is seen in the recovery of the ICN1 levels at 6 hours. This sentence has to be changed.

4) Again, the "dramatic decrease of NICD levels at 2h and 4h corresponding to CDK2-dependent G1/S phase" is perfectly detected in the CDK2 KOs, which does not match with author's interpretations (Notch should be stabilised in the KO?). These results have to be explained.

5) ICN1 staining in the embryos needs to be included.

During cross-commenting, referee #3 further suggested to included a table summarizing the delays in all inhibitor treated PSMs along with data indicating that bisecting and culturing both PSM halves in DMSO never causes a delay. To include the (appropriately repeated, quantified and statistically analysed) PSM western blot data in the body of the paper, and to clarify the analyses, data and interpretation of the somite length data as presented in figure 9.

As all three referees indicated that the study is very novel and of high general interest, we invite you to address these points in an exceptional further (and final) revision. Please address the remaining concerns (as detailed above) experimentally, or in a detailed point-by-point response. Please also address all the suggestions/points by referee #3 (see her/his report below) during revision.

Further, I have these editorial requests:

- Please add a running title (not more than 40 characters including spaces) to the title page in the main manuscript text file.

- Could you reduce the number of main figures (to not more than 8)? I think it would be no problem to fuse some of the present figures, or to combine/order the data differently to have less figure files. See also our guide for figure preparation:
http://www.embopress.org/sites/default/files/EMBOPress_Figure_Guidelines_061115.pdf

- As I mentioned in my previous decision letter, the Expanded View format, which will be displayed in the main HTML of the paper in a collapsible format, has replaced the Supplementary information. Thus, please select up to 5 images from the present Appendix file to be displayed as Expanded View (or include some data presently in the main figures). Please follow the nomenclature Figure EV1, Figure EV2 etc. The figure legend for these should be included in the main manuscript text file in a section called Expanded View Figure Legends after the main Figure Legends section. Thus, please select up to 5 figures as EV figures, name them accordingly, upload these as single files, and provide their legend in the main manuscript text. The remaining supplementary material should then be supplied in the Appendix file. For more details please refer also to our guide to authors:
<http://embor.embopress.org/authorguide#manuscriptpreparation>

- Please move the section 'Mathematical Models' to the Appendix. I think this does not need to be shown in the manuscript online. Interested readers could look up the model in the Appendix. Thus, please add this to the Appendix, and call it out accordingly in the main manuscript text. The tables and the figure should then be part of the Appendix Tables or Figures and labelled accordingly. Please use the nomenclature Appendix Figure Sx or Appendix Table Sx for these Appendix items, and change the callouts accordingly throughout the manuscript text.

- Please reformat the entire bar diagrams. X- and y-axes are hardly visible, or not visible at all. Also

the error bars are too thin, and partly not visible. Please also check that the asterisks are always visible (see e.g. the one in Fig. 5E that seems partly covered).

- The writing on the y-axes in Fig. 9E/F is too small. Please use bigger fonts. It would further be better to use the same style for all the bar diagrams.
- Regarding data quantification and statistics, please check for all diagrams that the number "n" for how many independent experiments (biological replicates) were performed and also the test used to calculate p-values is indicated in the respective figure legends. See: <http://embor.embopress.org/authorguide#statisticalanalysis>
- Could statistical testing be done (shown) for the diagrams in Fig. 9E/F and 10C?
- Was the raw data for the mass spec. deposited? If yes, please add the information (database, accession numbers) to the methods section.
- Please format the references according to EMBO reports style. Please use et al. only for references with more than 10 authors. In this case, the first 10 authors need to be listed. See: <http://embor.embopress.org/authorguide#referencesformat>
- There is an additional table in the methods section ('Oligonucleotide Sequences for siRNA Knockdown', page 34) that needs to be named as table. Please do that (table 2? - present Table 2 would then be Table 3). Both tables could also be moved to the Appendix.
- Please move the legends to the end of the main manuscript text file. Please also put the legend for Table 1 separately after the main figure legends.
- We now strongly encourage the publication of original source data with the aim of making primary data more accessible and transparent to the reader. The source data will be published in a separate source data file online along with the accepted manuscript and will be linked to the relevant figure. If you would like to use this opportunity, please submit the source data (for example scans of entire gels or blots, data points of graphs in an excel sheet, additional images, etc.) of your key experiments together with the revised manuscript. Please include size markers for scans of entire gels, label the scans with figure and panel number, and send one PDF file per figure.

- a Microsoft Word file (.doc) of the final revised manuscript text
- editable TIFF or EPS-formatted figure files (main figures and EV figures) in high resolution (of those with adjusted panels or labels).
- The revised Appendix.

REFEREE REPORTS

Referee #1:

The new version submitted by the authors is significantly improved, and they have already answered several of my previous questions. However, I really feel that authors have failed to neutralize my main concern that is the low amount of data dedicated to embryo segmentation, which is in fact replaced by data generated from HEK293 cells. Thus in my opinion, even when endogenous Notch1 is measured, HEK293 cells do not represent the best model for studying physiologic Notch function in the segmentation clock, as it is stated in the title, and this is not solved by including additional cellular models in specific experiments (randomly selected, apparently) along the manuscript. Moreover, although the authors have addressed some of the issues raised by the referees there are many controls lacking. For example, there is no evidence that the inhibitors do not affect full-length Notch1 thus leading to increased levels of the processed form. Related with this, Figure S2B does

not demonstrated that the effects of CK inhibition on Notch are ligand independent as the Notch-processing inhibitor (LY411575) is added only 1 hour before collecting the lysates but 2 hours after CK inhibitors treatment. One could argue that levels of NICN1 previous to LY incubation are already increased and the differences maintained 1 hour later. The only way to demonstrate that the effect of CDK inhibitors is on stability instead of processing is looking at protein decay in a time course experiment. Another misinterpretation of the data (in my opinion) is found in the WB analysis of synchronized HEK and CDK2 KO cells. There, it is stated that there is a dramatic decrease in fluctuations in the KO cells, whereas to my view the only difference is seen in the recovery of the ICN1 levels at 6 hours. However, the "dramatic decrease of NICD levels at 2h and 4h corresponding to CDK2-dependent G1/S phase" is perfectly detected in the CDK2 KOs, which does not match with author's interpretations (Notch should be stabilized in the KO?). Other examples where data is not adequately describe are in

Most importantly, questions/suggestions related with the embryo experiments have been primarily ignored. For example, two of the reviewers indicated the importance of showing Notch levels to support the conclusion that alterations in somitogenesis found in the CK-inhibited embryos (minor), are associated with changes in ICN1 stability (or at least levels), which is not addressed in the manuscript. The only piece of data regarding Notch levels in the treated embryos is a WB analysis in supplementary data that show minor changes in ICN1. Also Figure 9E and 9F are very difficult to interpret from the text but also from figure legends. Also, there is no reason to show *in vivo* data in the supplementary material instead of including as principal figures.

Referee #2:

This revised manuscript has been substantially improved by shortening text, moving a subset of figures to the Appendix and justifying the use of particular cell lines and methodologies in the study. Both *in vitro* and *in vivo* approaches are utilised to convincingly demonstrate regulation of NICD turnover by CDK1 and CDK2 and subsequent modulation of clock gene oscillation in somitogenesis.

The manuscript will be of interest to a wide readership including developmental biologists, those studying the Notch pathway and biological regulation by cyclin-dependent kinases.

Referee #3:

Overall, the authors have done an excellent job responding to reviewer concerns, and have largely mitigated my concerns. A few issues remain, that can likely be dealt with without further experimentation, as they largely regard interpretation and data presentation. My previous comments regarding novelty and importance stand, as I still feel that these results are interesting both to researchers studying the segmentation clock and to researchers interested in regulation of NICD turnover in a variety of contexts.

1. The author response to my point 2 about the phosphatase treatment still does not make sense to me. If the signal is maxed out, then the blot is not useful for quantification. Further, the basic question still remains for the quantification that includes all biological replicates. Why does treatment with phosphatase reduce the level of NICD seen after MLN treatment - there should be no change in levels, just in gel mobility, thus MLN+phosphatase treatment should have higher NICD levels than DMSO treated cells, and equivalent to cells treated with MLN alone. Or I'm fundamentally misunderstanding something which is not impossible.
2. In Fig 1B and D, (and S1B) why do the authors use the LY treated cells as the control for comparison? To make the claim that NICD levels are elevated after inhibitor treatment, the DMSO cells should be designated as the control for the Dunnett post-hoc test (the statistics as presented show that inhibitors increase NICD compared to a condition where NICD production is directly inhibited, which is not relevant to their model/conclusion).
3. In Figures 2B and D the values on the Y axis aren't compatible with fold change compared to DMSO which is what is indicated in the figure legend. In both cases, the values seem to be normalized to MLN treatment rather than to DMSO. Further the MLN value is expressed as 100%, which produces a percent change rather than a fold change.

4. Figure 8: Why are the G1 and G2 populations combined into a single group?
5. Figures S3B, S3D, and 9B are also expressed as percent rather than fold change (similar to point 3 above)
6. It would be interesting to know the effects of the CDK2 selective inhibitor on CDK1 and the CDK1 selective inhibitor on CDK2. Those values should be included (or referenced if already known)
7. The results of Figure 6 may be overinterpreted. The comparison of NICD levels between wild type and CDK2^{-/-} cells is qualitatively different in the experiment shown in 6A compared to 6C, and significantly different in 6B, but not in 6D, suggesting those effects are not especially robust. Further, when you compare NICD levels in wt cells treated with CDK1 siRNA to NICD levels in CDK2^{-/-} cells + CDK1 siRNA they look quite similar, which does not suggest that simultaneous loss of CDK1 and CDK2 has synergistic effects (or even additive effects), as stated on page 2 lines 336 and 349 and page 24, line 571
8. Figure 9 C/D. Please report the n for what fraction of RO-3306 treated half PSMs are delayed (similar to the n=10/18 reported for Purvalanol treated explants). It might also be useful to have a table indicating how delayed each explant counted as "delayed" was (one phase? 2? one somite behind) in both cases. If control experiments were done with both halves cultured in DMSO to confirm that delays are caused by the inhibitor, that should be reported as well.
9. Figure 9 E/F: Some additional explanation/interpretation of the results would be appreciated to explain why the data for somite index +1 different from those for somite index +2, and what that means in the context of this experiment and this model.

2nd Revision - authors' response

15 January 2019

We have responded to the comments you raise from the reviewers reports in the following way.

As you remember you said:

"... it would appear that referee #2 now supports the publication of the study in EMBO reports. Referee #3 has some remaining concerns s/he thinks could be fixed in a final revision most likely without extensive further experimentation. However, referee #1 has still major concerns. After cross-commenting, referee #1 indicates that minimally these points need to be addressed:

1) There is no evidence that CDK inhibitors do not affect full-length Notch1, subsequently leading to increased levels of the processed form. It is necessary to show levels of full-length Notch to clarify this issue. **We can tone down the conclusion in the discussion to say : while it is true we cannot formally exclude the possibility that the inhibitors may also affect full length Notch, it is nevertheless clear they have an effect on NICD stability since in the presence of LY411575, which inhibits Notch processing, we see an increase in levels of the processed form.**

2) Figure S2B does not demonstrated that the effects of CK inhibition on Notch are ligand independent as the Notch-processing inhibitor (LY411575) is added only 1 hour before collecting the lysates but 2 hours after CK inhibitors treatment. One could argue that levels of ICN1 previous to LY incubation are already increased and the differences maintained 1 hour later. The only way to demonstrate that the effect of CDK inhibitors is on stability instead of processing is looking at protein decay in a time course experiment.

We feel the reviewer may have missed the basis to our argument':

- **This is exactly what we are arguing – " that levels of ICN1 previous to LY incubation are already increased and the differences maintained 1 hour later"**
- **we report half-life of chicken NICD in the PSM is in the order of 11 mins (Wiedermann et al eLife 2015).**
- **thus as one might predict, 1h LY411575 treatment, (which prevents production/processing) leads to loss of NICD through degradation, as shown in DMSO lane.**
- **since we don't lose NICD in the inhibitor lanes the only conclusion is that that the CDK inhibitor treatment causes increased NICD stability.**

3) In the WB analysis of synchronized HEK and CDK2 KO cells, it is stated that there is a dramatic decrease in fluctuations in the KO cells when the only difference is seen in the recovery of the ICN1 levels at 6 hours. This sentence has to be changed. **We will lower the tone of this sentence as**

advised to state “we found a reduced effect upon fluctuations in NICD levels as compared to control cells”.

4) Again, the "dramatic decrease of NICD levels at 2h and 4h corresponding to CDK2-dependent G1/S phase" is perfectly detected in the CDK2 KOs, which does not match with author's interpretations (Notch should be stabilised in the KO?). These results have to be explained. We have explained this in lines 377-379 “Nevertheless some fluctuation was observed which may be attributable to functional compensation provided by CDK1 and associated cyclins' activity in the absence of CDK2 [77] [78]”.

5) ICN1 staining in the embryos needs to be included. We respectfully disagree, given that there would be no added value to this experiment, since *Lfng* is a target of Notch and therefore a readout of NICD activity in the PSM and these two markers have previously been shown to overlap in expression profile (Bone and Bailey et al Development 2014).

Referee #3 during cross-commenting further suggested to include a table summarizing the delays in all inhibitor treated PSMs along with data indicating that bisecting and culturing both PSM halves in DMSO never causes a delay. To include the (appropriately repeated, quantified and statistically analysed) PSM western blot data in the body of the paper, and to clarify the analyses, data and interpretation of the somite length data as presented in figure 9.

- We have provided a table as requested (table 2).
 - We have previously performed the suggested experiment and published that bisecting and culturing both PSM halves in DMSO does not cause a delay.
 - We have responded to this request - to move the PSM western blot data in embryos into the body of the paper – the reason it was in supplementary is that, in response to the first round of revisions we were asked to simplify the main body of the paper and move some data to the supplementary figures.
- We have included some additional explanation/interpretation of the results to explain why the data for somite index +1 different from those for somite index +2, and what that means in the context of this experiment and this model: We apologise as there may be some confusion regarding the notation for somite index which we can explain. By +1 we mean the most recently formed somite. This corresponds on average to four somites after the drug was added. By +2 we mean the penultimate somite. This corresponds on average to three somites after the drug was added. These results are consistent with the current understanding that the position of the somite boundaries is already specified for the first few somites that will form from the anterior region of the PSM (Dubrulle et al 2001), and thus for those that form just after the time when the drug was added. Our data suggest that somite boundary position has been specified at +1, +2, +3 but not +4.

3rd Editorial Decision

28 January 2019

Thank you for the submission of your revised manuscript to our editorial offices. We have now received the reports from two of the referees that were asked to re-evaluate your study, you will find below. As you will see, referee #1 now supports the publication of your manuscript in EMBO reports. However, referee #3 indicates that several of her/his comments have not been addressed. Indeed I had asked you to also address all the points by referee #3 during revision. It seems none of these have been addressed, as they are also not mentioned in the point-by-point response. I assume that this was an oversight, and therefore ask you to address these (also re-iterated in the new report of referee #3) in a further revised manuscript. Please find these points at the end of this message.

Further, there are these editorial requests that need attention:

- We require that large-scale datasets, sequences, atomic coordinates and computational models should be deposited in one of the relevant public databases. Please do that for your mass spec. data, and provide the accession codes in the final revised manuscript. See also: <http://embor.embo.org/authorguide#datadeposition>

- Please add scale bars to the microscopic panels in Fig. 6A, 7E/F and S3F. Please add simple bars without any writing to the figure, and define the length in the respective figure legend.

REFEREE REPORTS

Referee #1:

Although I feel that authors have not included any additional change to their answers to reviewer 1, I decline imposing an additional round of revisions if reviewers 2 and 3 are happy with the content of the manuscript.

Referee #3:

Overall, the authors have done an excellent job regarding the reviewer and editorial concerns. The manuscript is clearer and easier to follow and does a nice job discussing the importance of NICD turnover in a number of contexts. That said, there are a few trailing concerns left from my previous review that I believe the authors may have overlooked, perhaps due to confusion about which elements of the reviews needed to be addressed. Two of these I feel are critical:

1) In Fig 1B and D, and (now) EV1B, the labeling on the figures and the figure legend indicate that authors used the level of NICD in the LY411575 treated cells as the control for statistical comparison. This statistical analysis indicates only that NICD levels are higher in all cell lines that make NICD than in the cell line where gamma secretase is inhibited and NICD production is prevented. Although this is no doubt true, it is not relevant to the model/conclusion they draw from the data.

To support the claim that NICD levels are elevated after inhibitor treatment, the DMSO cells should be designated as the control for the Dunnett post-hoc test. Looking at the data, it seems likely to me that with this analysis, the GSI treated NICD levels will be statistically lower than the control, and most of the inhibitor treatments will result in statistically significant increases in NICD levels. Thus, this just requires the authors to re-run the statistics and relabel the figure, but they must do appropriate statistics to support their claims.

2) Although the data have been moved to the supplement (EV2 C-D), the authors still don't have data that supports the claim that simultaneous loss of CDK1 and CDK2 has a synergistic effect on NICD levels.

When you compare NICD levels in wt cells treated with CDK1 siRNA (column 2) to NICD levels in CDK2^{-/-} cells + CDK1 siRNA (column 4, functionally knocking down both CDK1 and 2) the NICD levels look quite similar. This does not suggest that simultaneous loss of CDK1 and CDK2 has synergistic effects (or even additive effects) in NICD levels. In order to make a claim of additive or synergistic effects, the levels of NICD after simultaneous loss of CDK1 and 2 would need to be higher than seen with the loss of just CDK1. This is not the case in the data presented by the authors. The data as presented are consistent with CDK1 and CDK2 having redundant or overlapping effects in the assay shown. The use of synergistic in the results and discussion should be altered.

As a minor point, I also note that in Figures 2B and D the values on the Y axis aren't compatible with fold change compared to DMSO which is what is indicated in the figure legend. In both cases, the values seem to be normalized to MLN treatment rather than to DMSO. Further the MLN value is expressed as 100%, which produces a percent change rather than a fold change. Similar notes apply to Figures 7B. The authors may wish to change the language in the legend to match the data presentation.

Referee #3 (previous version - not addressed):

Overall, the authors have done an excellent job responding to reviewer concerns, and have largely mitigated my concerns. A few issues remain, that can likely be dealt with without further experimentation, as they largely regard interpretation and data presentation. My previous comments regarding novelty and importance stand, as I still feel that these results are interesting both to researchers studying the segmentation clock and to researchers interested in regulation of NICD turnover in a variety of contexts.

1. The author response to my point 2 about the phosphatase treatment still does not make sense to me. If the signal is maxed out, then the blot is not useful for quantification. Further, the basic question still remains for the quantification that includes all biological replicates. Why does treatment with phosphatase reduce the level of NICD seen after MLN treatment - there should be no change in levels, just in gel mobility, thus MLN+phosphatase treatment should have higher NICD levels than DMSO treated cells, and equivalent to cells treated with MLN alone. Or I'm fundamentally misunderstanding something, which is not impossible.

2. In Fig 1B and D, (and S1B) why do the authors use the LY treated cells as the control for comparison? To make the claim that NICD levels are elevated after inhibitor treatment, the DMSO cells should be designated as the control for the Dunnett post-hoc test (the statistics as presented show that inhibitors increase NICD compared to a condition where NICD production is directly inhibited, which is not relevant to their model/conclusion).

3. In Figures 2B and D the values on the Y-axis aren't compatible with fold change compared to DMSO which is what is indicated in the figure legend. In both cases, the values seem to be normalized to MLN treatment rather than to DMSO. Further the MLN value is expressed as 100%, which produces a percent change rather than a fold change.

4. Figure 8: Why are the G1 and G2 populations combined into a single group?

5. Figures S3B, S3D, and 9B are also expressed as percent rather than fold change (similar to point 3 above).

6. It would be interesting to know the effects of the CDK2 selective inhibitor on CDK1 and the CDK1 selective inhibitor on CDK2. Those values should be included (or referenced if already known).

7. The results of Figure 6 may be overinterpreted. The comparison of NICD levels between wild type and CDK2^{-/-} cells is qualitatively different in the experiment shown in 6A compared to 6C, and significantly different in 6B, but not in 6D, suggesting those effects are not especially robust. Further, when you compare NICD levels in wt cells treated with CDK1 siRNA to NICD levels in CDK2^{-/-} cells + CDK1 siRNA they look quite similar, which does not suggest that simultaneous loss of CDK1 and CDK2 has synergistic effects (or even additive effects), as stated on page 2 lines 336 and 349 and page 24, line 571.

8. Figure 9 C/D. Please report the n for what fraction of RO-3306 treated half PSMs are delayed (similar to the n=10/18 reported for Purvalanol treated explants). It might also be useful to have a table indicating how delayed each explant counted as "delayed" was (one phase? 2? one somite behind) in both cases. If control experiments were done with both halves cultured in DMSO to confirm that delays are caused by the inhibitor that should be reported as well.

9. Figure 9 E/F: Some additional explanation/interpretation of the results would be appreciated to explain why the data for somite index +1 different from those for somite index +2, and what that means in the context of this experiment and this model.

Referee #3:

1) In Fig 1B and D, and (now) EV1B, the labeling on the figures and the figure legend indicate that authors used the level of NICD in the LY411575 treated cells as the control for statistical comparison. This statistical analysis indicates only that NICD levels are higher in all cell lines that make NICD than in the cell line where gamma secretase is inhibited and NICD production is prevented. Although this is no doubt true, it is not relevant to the model/conclusion they draw from the data. To support the claim that NICD levels are elevated after inhibitor treatment, the DMSO cells should be designated as the control for the Dunnett post-hoc test. Looking at the data, it seems likely to me that with this analysis, the GSI treated NICD levels will be statistically lower than the control, and most of the inhibitor treatments will result in statistically significant increases in NICD levels. Thus, this just requires the authors to re-run the statistics and relabel the figure, but they must do appropriate statistics to support their claims.

We performed the comparison using DMSO in the first version of the manuscript. We then decided to use LY because in some cases in DMSO is not easy to detect NICD because of its fast turnover. In this version of the manuscript, we have reverted to compare NICD levels following drug treatments as compared to DMSO as we agree with the reviewers comments.

2) Although the data have been moved to the supplement (EV2 C-D), the authors still don't have data that supports the claim that simultaneous loss of CDK1 and CDK2 has a synergistic effect on NICD levels. When you compare NICD levels in wt cells treated with CDK1 siRNA (column 2) to NICD levels in CDK2^{-/-} cells + CDK1 siRNA (column 4, functionally knocking down both CDK1 and 2) the NICD levels look quite similar. This does not suggest that simultaneous loss of CDK1 and CDK2 has synergistic effects (or even additive effects) in NICD levels. In order to make a claim of additive or synergistic effects, the levels of NICD after simultaneous loss of CDK1 and 2 would need to be higher than seen with the loss of just CDK1. This is not the case in the data presented by the authors. The data as presented are consistent with CDK1 and CDK2 having redundant or overlapping effects in the assay shown. The use of synergistic in the results and discussion should be altered.

We thank the reviewer for this comment. However we would like to suggest that the correct control to look at loss of CDK1 in addition to loss of CDK2 has to be done in the CDK2^{-/-} background - this is exactly what we do show in Figure EV2C comparing levels of NICD in lanes 5 and 6 with levels in lanes 7 and 8 and in the quants in Figure EV2 D).

As a minor point, I also note that in Figures 2B and D the values on the Y axis aren't compatible with fold change compared to DMSO which is what is indicated in the figure legend. In both cases, the values seem to be normalized to MLN treatment rather than to DMSO. Further the MLN value is expressed as 100%, which produces a percent change rather than a fold change. Similar notes apply to Figures 7B. The authors may wish to change the language in the legend to match the data presentation.

We have changed the legends accordingly as suggested.

----- Referee #3 (previous version - not addressed):

1. The author response to my point 2 about the phosphatase treatment still does not make sense to me. If the signal is maxed out, then the blot is not useful for quantification. Further, the basic question still remains for the quantification that includes all biological replicates. Why does treatment with phosphatase reduce the level of NICD seen after MLN treatment - there should be no change in levels, just in gel mobility, thus MLN+phosphatase treatment should have higher NICD levels than DMSO treated cells, and equivalent to cells treated with MLN alone. Or I'm fundamentally misunderstanding something, which is not impossible.

We agree with the reviewers' comments and the data suggests an additional scientific possibility, which is that when **all** phosphorylation modifications are removed by lambda phosphatase treatment then NICD is less stable and degraded by a mechanism other than SCF E3 ligase. We have added this observation to the main text.

2. In Fig 1B and D, (and S1B) why do the authors use the LY treated cells as the control for comparison? To make the claim that NICD levels are elevated after inhibitor treatment, the DMSO cells should be designated as the control for the Dunnett post-hoc test (the statistics as presented show that inhibitors increase NICD compared to a condition where NICD production is directly inhibited, which is not relevant to their model/conclusion).
See comment above.

3. In Figures 2B and D the values on the Y-axis aren't compatible with fold change compared to DMSO which is what is indicated in the figure legend. In both cases, the values seem to be normalized to MLN treatment rather than to DMSO. Further the MLN value is expressed as 100%, which produces a percent change rather than a fold change.
We have rectified this in the legend.

4. *Figure 8: Why are the G1 and G2 populations combined into a single group?*

We specifically wanted to monitor levels of NICD where CDK activity peaks. Furthermore, we were interested in whether those two phases had higher NICD levels than M and S phase, so we did the comparison but combined the median fluorescence intensity (MFI) for G1 and G2 cells.

5. *Figures S3B, S3D, and 9B are also expressed as percent rather than fold change (similar to point 3 above).*

We have rectified this as requested.

6. *It would be interesting to know the effects of the CDK2 selective inhibitor on CDK1 and the CDK1 selective inhibitor on CDK2. Those values should be included (or referenced if already known).*

We agree with the referee; however, this data is not available from the MRC database.

7. *The results of Figure 6 may be overinterpreted. The comparison of NICD levels between wild type and CDK2^{-/-} cells is qualitatively different in the experiment shown in 6A compared to 6C, and significantly different in 6B, but not in 6D, suggesting those effects are not especially robust. Further, when you compare NICD levels in wt cells treated with CDK1 siRNA to NICD levels in CDK2^{-/-} cells + CDK1 siRNA they look quite similar, which does not suggest that simultaneous loss of CDK1 and CDK2 has synergistic effects (or even additive effects), as stated on page 2 lines 336 and 349 and page 24, line 571.*

We have responded to this comment in point 2 above.

8. *Figure 9 C/D. Please report the n for what fraction of RO-3306 treated half PSMs are delayed (similar to the n=10/18 reported for Purvalanol treated explants). It might also be useful to have a table indicating how delayed each explant counted as "delayed" was (one phase? 2? one somite behind) in both cases. If control experiments were done with both halves cultured in DMSO to confirm that delays are caused by the inhibitor that should be reported as well.*

We have already responded in full to this comment in the 2nd re-submitted version by including a table as requested highlighting how delayed each explant is and we have previously published the control experiment with both halves cultured in DMSO.

9. *Figure 9 E/F: Some additional explanation/interpretation of the results would be appreciated to explain why the data for somite index +1 different from those for somite index +2, and what that means in the context of this experiment and this model.*

We have already responded to this comment in the re-submitted version as follows: we have included some additional explanation/interpretation of the results to explain why the data for somite index +1 different from those for somite index +2, and what that means in the context of this experiment and this model: We apologise as there may be some confusion regarding the notation for somite index which we can explain. By +1 we mean the most recently formed somite. This corresponds on average to four somites after the drug was added. By +2 we mean the penultimate somite. This corresponds on average to three somites after the drug was added. These results are consistent with the current understanding that the position of the somite boundaries is already specified for the first few somites that will form from the anterior region of the PSM (Dubrulle et al 2001), and thus for those that form just after the time when the drug was added. Our data suggest that somite boundary position has been specified at +1, +2, +3 but not +4.

Accepted

25 March 2019

Thanks for the submission of the final revised version of your manuscript. I have now received the report of from referee #3 that was asked to re-evaluate your study, which can be found below.

As you will see, the referee now supports the publication of your study. I am thus very pleased to accept your manuscript for publication in the next available issue of EMBO reports. Thank you for your contribution to our journal.

At the end of this email I include important information about how to proceed. Please ensure that you take the time to read the information and complete and return the necessary forms to allow us to publish your manuscript as quickly as possible.

As part of the EMBO publication's Transparent Editorial Process, EMBO reports publishes online a Review Process File to accompany accepted manuscripts. As you are aware, this File will be published in conjunction with your paper and will include the referee reports, your point-by-point response and all pertinent correspondence relating to the manuscript.

If you do NOT want this File to be published, please inform the editorial office within 2 days, if you have not done so already, otherwise the File will be published by default [contact: emboreports@embo.org]. If you do opt out, the Review Process File link will point to the following statement: "No Review Process File is available with this article, as the authors have chosen not to make the review process public in this case."

Should you be planning a Press Release on your article, please get in contact with emboreports@wiley.com as early as possible, in order to coordinate publication and release dates.

Thank you again for your contribution to EMBO reports and congratulations on a successful publication. Please consider us again in the future for your most exciting work.

REFEREE REPORT

Referee #3:

The authors have responded to all reviewer requests, producing a strong and exciting manuscript.

I still fundamentally disagree with their claim of synergistic effects of simultaneous loss of cdk1 and cdk2. I think of synergy using the definition here : <https://www.ncbi.nlm.nih.gov/pubmed/19665253> that "Synergy occurs when the contribution of two mutations to the phenotype of a double mutant exceeds the expectations from the additive effects of the individual mutations." Given that definition, looking at NICD levels in the figure with a 3 fold increase from CDK1 siRNA and about a 1.5 fold increase in cdk2 null cells, a claim of synergy would require a greater than 3.5 fold increase in the cdk2 null cells treated with cdk1 siRNAs.

That said, I'm not going to insist that the authors accept this definition, and am happy to suggest acceptance of this interesting manuscript.

Corresponding Author Name: J. Kim Dale

Journal Submitted to: EMBO reports

Manuscript Number: EMBOR-2018-46436V2